# Asymptotic theory of SGD with a general learning-rate

**Or Goldreich**
Department of Statistics
University of Chicago
Chicago, IL 60637
orgoldreich@uchicago.edu

**Ziyang Wei**
Department of Statistics
University of Chicago
Chicago, IL 60637
ziyangw@uchicago.edu

**Soham Bonnerjee**
Department of Statistics
University of Chicago
Chicago, IL 60637
sohambonnerjee@uchicago.edu

**Jiaqi Li**
Department of Statistics
University of Chicago
Chicago, IL 60637
jqli@uchicago.edu

**Wei Biao Wu**
Department of Statistics
University of Chicago
Chicago, IL 60637
wbwu@chicago.edu

## Abstract

Stochastic gradient descent (SGD) with polynomially decaying step-sizes has long underpinned theoretical analyses, yielding a broad spectrum of statistically attractive guarantees. Yet in practice, such schedules find rare use due to their prohibitively slow convergence, revealing a persistent gap between theory and empirical performance. In this paper, we introduce a unified framework that quantifies the uncertainty of online SGD under arbitrary learning-rate choices. In particular, we provide the first comprehensive convergence characterizations for two widely used but theoretically under-examined schemes—cyclical learning rates and linear decay to zero. Our results not only explain the observed behavior of these schedules but also facilitate principled tools for statistical inference and algorithm design. All theoretical findings are corroborated by extensive simulations across diverse settings.

## 1 Introduction

Stochastic Gradient Descent (SGD) has gained popularity in modern machine learning since the seminal work of Robbins and Monro [1951]. While its theoretical foundations are well established, the literature has largely focused on two standard step-size choices: *constant step-sizes*, which provide exponentially fast convergence to a biased stationary distribution and allow straightforward tuning, and *polynomially decaying step-sizes*, typically of the form $\eta_t \asymp t^{-\alpha}$, which offer statistical guarantees such as consistency and asymptotic normality [Chung, 1954, Sacks, 1958, Fabian, 1968, Ruppert, 1988, Polyak and Juditsky, 1992] and extend to many SGD variants [Poljak, 1964, Gadat and Panloup, 2023, Li et al., 2024b]. However, polynomially decaying schedules converge slowly in practice, while constant step-sizes require careful calibration to avoid divergence [Bengio, 2012]. Hybrid schemes combining these approaches have gained traction in deep learning [He et al., 2016, Smith and Topin, 2019], though learning rates are often chosen empirically or via hyper-parameter tuning over standard schedules [Wu et al., 2018]. This practical reliance leaves a notable gap in theoretical understanding of general step-size effects on SGD. The critical role of learning rates in stochastic approximation convergence has long been recognized [Spall, 2003, Nemirovski et al., 2008], underscoring the need for a unified theoretical framework encompassing a broader range of step-size strategies.

Recent applications have introduced a variety of learning rate schedules that, despite their empirical success, lack comprehensive theoretical support. For instance, Smith [2017] proposes several cyclical

learning rate schemes that perform well in practice across standard neural network architectures. Another under-theorized category is that of finite-horizon schedules, where the learning rate depends explicitly on the total number of iterations. In high-dimensional linear regression, for example, Agrawalla et al. [2023] recommends a schedule of the form $\eta_{t,n} \propto \frac{\log n}{n}$. Among such schedules, the linearly decaying to zero (Linear-D2Z) learning rate has seen widespread use in training large-scale architectures. Detailed discussion on the relevant literature – though by no means exhaustive – is included in Section 1.3. Despite its prevalence, the non-asymptotic behavior of Linear-D2Z and related finite-horizon step-size policies remains poorly understood from a theoretical standpoint. In this work, we aim to bridge this theoretical gap by (i) developing a unified framework to show the non-asymptotic moment convergence, and (ii) explicitly characterizing the non-asymptotic behavior of the wide class of cyclical learning rates as well as Linear-D2Z.

## 1.1 SGD preliminaries

Before summarizing the main contributions of this article, we briefly introduce the stochastic gradient descent (SGD) problem and establish a consistent notational framework for the analysis. Consider the problem of minimizing a function $F : \mathbb{R}^d \to \mathbb{R}$, $F \in \mathcal{C}^1$, given by:

$$\theta^\star = \arg\min_{\theta \in \mathbb{R}^d} F(\theta),$$

and the corresponding SGD algorithm:

$$\theta_t = \theta_{t-1} - \eta_t \nabla f(\theta_{t-1}, \xi_t), \quad \theta_0 \in \mathbb{R}^d, \tag{1.1}$$

where $\eta_t$ are step-sizes at $t$-th step, and $\xi_1, \xi_2, \ldots$, are i.i.d. samples from some unknown distribution $\mathbb{P}_\xi$ such that $\mathbb{E}_{\xi \sim \mathbb{P}_\xi}[\nabla f(\theta_{t-1}, \xi_t)] = \nabla F(\theta)$ for all $\theta \in \mathbb{R}^d$. With this formulation in place, we now proceed to outlining the main contributions of this work.

## 1.2 Main contributions

This article presents a unified framework for deriving the non-asymptotic mean-squared error for arbitrary learning rate schedules. Our results not only encompass the known theory for polynomially decaying learning rates as a specific case, but also extend the asymptotic analysis to other commonly used learning rates that previously lacked theoretical support. Our main contributions are summarized below.

**(1)** In our Theorem 2.1, we prove a general bound on the mean error of the SGD iterates, $\mathbb{E}[|\theta_n - \theta^\star|^p]$ involving the step sizes $\{\eta_t\}$. In particular, under certain regularity conditions, we prove the following.

**Theorem 1.1** (Theorem 2.1, informal). *If $S_{s,t} = \sum_{j=s+1}^t \eta_j$ for $t > s$, then it holds that*

$$\mathbb{E}[|\theta_n - \theta^\star|^p] \lesssim \exp(-c_p S_{1,n})|\theta_0 - \theta^\star|^p + \sum_{j=1}^n \eta_j^2 \exp(-c_p S_{j,n}).$$

This result enables straightforward application to a wide range of learning rate schedules $\eta_t$, provided that the second term, $\sum_{j=1}^n \eta_j^2 \exp(-c_p S_{j,n})$, can be effectively controlled. For example, such bounds are typically tractable for many "approximately" polynomially decaying learning rates. Moreover, the result explicitly captures the influence of the initial point on the final error of the SGD iterates.

**(2)** A key aspect of Theorem 2.1 is that the step sizes $\eta_j$ are allowed to depend on the total iteration count $n$, thereby accommodating finite-horizon learning rate schedules. Such schedules have rarely been studied in the context of mean-squared error. We specifically examine the widely used linearly decaying schedule $\eta_t = \eta(1 - t/n)$ as a representative case. In Theorem 2.2, we leverage Theorem 2.1 to characterize the non-asymptotic moment convergence of SGD iterates under this step-size rule. Although this schedule is common in practice, Theorem 2.2 provides, to the best of our knowledge, its first explicit, rigorous analysis in the online SGD setting. Furthermore, our approach can be readily extended to a broad class of finite-horizon schedules.

**(3)** When the learning rate is constant ($\eta_t \equiv \eta$) or cyclical ($\eta_t = \eta_{t \bmod T}$), Theorem 2.1 yields only an $O(1)$ bound, which offers only limited insight. While it is well established that, in the constant

case, the SGD iterates converge to a stationary distribution, there is, to the best of our knowledge, no existing asymptotic theory for the cyclical learning rate setting. In Theorem 2.4, we address this gap by presenting a novel convergence result for SGD iterates under cyclical learning rate schedules.

**Theorem 1.2** (Theorem 2.4, informal). *For a cyclical learning rate $\boldsymbol{\eta} = (\eta_1, \ldots, \eta_T)$, if not all of the $\eta_k$'s are too big, then there exists a "cyclostationary" process $\pi$ such that*

$$\theta_n \overset{w}{\Rightarrow} \pi, \ as \ n \to \infty,$$

*where $\overset{w}{\Rightarrow}$ denotes the convergence in distribution.*

This result highlights a fundamental behavioral difference between SGD with cyclical learning rates and with constant learning rates. While the latter typically converges to a stationary distribution, resembling the behavior of a Markov chain, the former converges to a distinct type of non-stationary distribution exhibiting periodic patterns over time- formally known as cyclostationary distribution. To aid the reader's understanding, we also include a brief but formal discussion of cyclostationary processes.

**(4)** Our theoretical results are substantiated by extensive numerical exercises. Section 3.2 focuses on linearly decaying schedules, which empirically demonstrate both fast early convergence and low final error, consistent with Theorem 2.2, and justifying their practical appeal. On the other hand, Section 3.3 examines cosine schedules, where the learning rate follows a smooth periodic pattern; the resulting error exhibits cyclical behavior, particularly in the variance of its estimate, also complimenting Theorem 2.4. Some additional numerical exercises can be found in Appendix C. Specifically, Appendix C.6 collects five different learning schedules, and provides a comparative study that highlights both their behavior in the "transient" (i.e. with respect to initialization) phase, as well as their asymptotic behavior.

## 1.3 Related works

There exists a substantial, though primarily empirical, body of literature examining gradient descent and batched SGD in the context of neural networks. For instance, Wu et al. [2019] investigates a variety of step-size schedules, including exponentially decaying and time-inverted schemes. Among the many proposed strategies, our focus is on two broad and widely used classes: cyclical schedules and linearly decaying schedules. Since the introduction of the "triangular" learning rate by Smith [2017], periodic learning rate schemes—and their decaying variants such as cosine annealing—have become influential in training deep architectures like convolutional neural networks (CNNs) [Loshchilov and Hutter, 2017, Smith, 2023, Wang et al., 2023]. The periodic structure of these schedules allows for intermittent large steps (which encourage exploration) followed by smaller steps (which promote convergence), a behavior associated with so-called "super-convergence" as observed in both empirical and theoretical work Smith and Topin [2019], Oymak [2021].

In parallel, annealing-based strategies have also played a prominent role in optimization [Huang et al., 2017, Li et al., 2019, Nakkiran, 2020], with certain variants—such as geometrically decaying step-sizes—proven to be minimax optimal in convex settings [Ge et al., 2019]. Within this context, the linearly decaying to zero (Linear-D2Z) schedule has gained significant traction in applications involving highly non-smooth or complex optimization landscapes, including state-space models [Touvron et al., 2023], large language models [Devlin et al., 2019, Liu et al., 2019, Bergsma et al., 2025], and vision transformers [Wu et al., 2024]. Notably, several works advocate for a "knee schedule" [Howard and Ruder, 2018, Hoffmann et al., 2022, Iyer et al., 2023, Defazio et al., 2023, Hägele et al., 2024, Bergsma et al., 2025], which begins with a large learning rate (a "warm start") followed by a Linear-D2Z phase. Despite their widespread adoption, the asymptotic behavior of both cyclical and Linear-D2Z step-size schedules remains theoretically unexplored—even in relatively simple convex settings. This lack of theoretical understanding presents a significant barrier to rigorous statistical inference and uncertainty quantification, underscoring the need for systematic analysis.

## 1.4 Notations

In this paper, we denote the set $\{1, \ldots, n\}$ by $[n]$. The $d$-dimensional Euclidean space is $\mathbb{R}^d$. For a vector $a \in \mathbb{R}^d$, $|a|$ denotes its Euclidean norm. For a random vector $X \in \mathbb{R}^d$ and $s > 0$, we

denote $\|X\| := \sqrt{\mathbb{E}[|X|^2]}$ and $\|X\|_s = (\mathbb{E}[|X|^s])^{1/s}$. We also denote in-probability convergence, and stochastic boundedness by $o_{\mathbb{P}}$ and $O_{\mathbb{P}}$ respectively. We write $a_n \lesssim b_n$ if $a_n \leq C b_n$ for some constant $C > 0$, and $a_n \asymp b_n$ if $C_1 b_n \leq a_n \leq C_2 b_n$ for some constants $C_1, C_2 > 0$. Often we denote $a_n \lesssim b_n$ by $a_n = O(b_n)$. Additionally, if $a_n/b_n \to 0$, we write $a_n = o(b_n)$.

## 2 Non-asymptotic moment convergence of SGD iterates with general step-sizes

This section is devoted to establishing the $p$-th moment convergence of SGD iterates (1.1) for any $p \geq 2$ with a general choice of learning rate. In particular, we allow for finite-horizon schedules; in the notation of Section 1.1, we allow $\eta_t \equiv \eta_{t,n}$. We note that this represents a significant improvement the existing body of literature that analyzes the statistical properties of SGD and its variants under different learning rate schedules. Before we discuss our main result, it is imperative to introduce the crucial technical assumptions behind our result.

### 2.1 Technical assumptions

We assume the following regularity assumptions.

**Assumption 2.1.** *The function $F$ is $\mu$-strongly convex, i.e. for a $\mu > 0$ and for all $x, y \in \mathbb{R}^d$, it holds that*
$$\langle \nabla F(x) - \nabla F(y), x - y \rangle \geq \mu |x - y|^2.$$

The strong-convexity assumption 2.1 can further be relaxed into the strong concordance assumption as follows:

**Assumption 2.2** (Local strong concordance). *There exists $\mu^\star > 0$ such that $\nabla_2 F(\theta^\star) \succeq \mu^\star \boldsymbol{I}_d$. Moreover, there exists a constant $C > 0$, and compact set $\Phi \subseteq \mathbb{R}^d$, such that for all $\theta_1, \theta_2 \in \Phi$, it holds that*
$$|\varphi'''(u)| \leq C |\theta_1 - \theta_2| \, \varphi''(u), \text{ where } \varphi : u \mapsto F(\theta_1 + u(\theta_2 - \theta_1)), u \in \mathbb{R}.$$

We remark that adoption of Assumption of 2.2 instead of Assumption 2.1 does not significantly alter any of our arguments; see Gu and Chen [2024] for details. For simplicity, we stick with Assumption 2.1.

**Assumption 2.3.** *For the noisy gradients $\nabla f(\cdot, \cdot)$ and some $p \geq 2$, there exists a constant $L_p > 0$ such that*
$$\mathbb{E}[|\nabla f(x, \xi) - \nabla f(y, \xi)|^p] \leq L_p^p |x - y|^p, \quad \text{for all } x, y \in \mathbb{R}^d.$$
*In particular, for some constant $M_p > 0$, it holds that*
$$(\mathbb{E}[|\nabla f(\theta^\star, \xi)|^p])^{1/p} =: M_p < \infty.$$

Assumption 2.3 entails that $F$ is $L_p$-smooth by Hölder's inequality; in other words, for all $x, y \in \mathbb{R}^d$, it holds that
$$|\nabla F(x) - \nabla F(y)| \leq L_p |x - y|.$$

Assumptions 2.1 and 2.3 are standard features of statistical analysis of convex stochastic optimization, and have appeared extensively in Ruppert [1988], Polyak and Juditsky [1992], Bottou et al. [2018], Chen et al. [2020], Zhu et al. [2023], Wei et al. [2023], Li et al. [2024a]. With these standard regularity assumptions, we can introduce our general result.

### 2.2 A general moment convergence of SGD iterates

In this section, we introduce our main contribution – an umbrella result that furnishes a ready-made upper-bound of the SGD iterates (1.1) for any choice of learning rates. In particular, we have the following result.

**Theorem 2.1** ($L^p$ Convergence). *Suppose that Assumptions 2.1 and 2.3 hold for some $p \geq 2$. Let $c_0 > 0$ be some constant such that for all $t \geq 1$, $c_0 \leq \min\left\{\eta_t^{-1}, 2\mu - (6p - 5)L_p^2 \eta_t\right\}$. For the learning rate schedule $\eta_t$ satisfies*
$$0 < \eta_t < \frac{2\mu}{(6p - 5)L_p^2}, \tag{2.1}$$

*we have for any $n \geq 1$,*

$$\|\theta_n - \theta^\star\|_p^2 \leq \exp\left\{ - c_0 \sum_{k=1}^{n} \eta_k \right\}|\theta_0 - \theta^\star|^2 + 3(p-1)M_p^2 \sum_{j=1}^{n} \eta_j^2 \exp\left\{ - c_0 \sum_{k=j+1}^{n} \eta_k \right\}. \quad (2.2)$$

Theorem 2.1 is proved in appendix Section A.1. The bound (2.2) highlights the two key terms that the learning rates contribute in the moment bound. In particular, there is an inherent trade-off between the potential choices of step-sizes $\eta_t$ that goes into determining the order of the $p$-th moment. We discuss this property in detail in the subsequent two remarks.

*Remark* 2.1 (Effect of initialization). Firstly, the $\exp$ term in (2.2) highlights that in order to neglect the effect of initialization, one must have $\sum_{k=1}^{n} \eta_k \to \infty$ as $n \to \infty$; in other words, the step-sizes cannot be too small. For example, Wu et al. [2019] discusses exponentially decaying step-sizes $\eta_t = \gamma^t$, whose performance heavily depends on the initial point even for large $n$, indicating that the effect of initialization cannot be ignored in this case.

*Remark* 2.2 (Effect of exploration). The second term $\sum_{j=1}^{n} \eta_j^2 \exp\left\{ - c_0 \sum_{k=j+1}^{n} \eta_k \right\}$ encodes the exploration property of the SGD iterates $\theta_t$. Intuitively, if $\eta_n \to 0$ as $n \to \infty$, then this second term is also $o(1)$. Therefore, this term essentially ensures that $\eta_j$ has to be decaying, and not all of them can be too big.

It is instructive to examine specific learning rate choices and their implications as reflected by the bound in (2.2). For example, with the commonly used polynomially decaying schedule $\eta_t \asymp t^{-\beta}$, the first term behaves like $\exp(-t^{1-\beta})$, while the second term is on the order of $O(\eta_t)$, recovering the classical mean square error (MSE) rate for this setting. In the following section, we apply Theorem 2.1 to analyze another important and theoretically less-explored finite-horizon schedule: the linearly decaying to zero (`Linear-D2Z`) learning rate.

## 2.3 Linear decaying step-sizes

As an important application of Theorem 2.1, consider the `Linear-D2Z` learning rate $\eta_t = \eta(1 - t/n)$. This learning schedule has recently been at the forefront of training large architectures, and its optimality properties have been investigated both theoretically [Defazio et al., 2023] and empirically Bergsma et al. [2025] in different context. Despite this interest, its non-asymptotic convergence rate remains unknown in the literature. Leveraging the bound in (2.2), we analyze the $L_p$ convergence behavior of SGD under this learning rate schedule.

**Theorem 2.2.** *Recall $\theta_n$ from* (1.1). *Under the conditions of Theorem 2.1, we have*

$$\|\theta_n - \theta^\star\|_p^2 \leq |\theta_0 - \theta^\star|^2 \exp\{-\frac{c_0 \eta(n-1)}{2}\} + \frac{C}{\sqrt{n}},$$

*where $C > 0$ is a universal constant independent with $n$ and $\theta_0$.*

*Remark* 2.3. Theorem 2.2 offers a remarkable insight into the behavior of the linearly decaying learning rate: it effectively combines the advantages of both constant and polynomially decaying step-sizes by being consistent and forgetting the initial condition at an exponential rate. Specifically, for any $c \in (0, 1)$ and all iterations $t \leq \lfloor nc \rfloor$, the step size satisfies $\eta_t \geq \eta(1-c)$; that is, the learning rate behaves like a constant for a substantial portion of the optimization process, providing a "warm start" and ensuring exponential decay relative to the initial point. Conversely, when $t = \lceil n - c_0 n^{1-c} \rceil$ for some $c > 1/2$ and $c_0 > 0$, the step size satisfies $\eta_t = \eta c_0 n^{-c} \asymp t^{-c}$, mimicking a polynomially decaying schedule that yields the MSE of order $O(n^{-1/2})$. Therefore, by leveraging the strengths of both constant and polynomially decaying learning rates, the linearly decaying to zero (`Linear-D2Z`) schedule achieves a "best-of-both-worlds" effect. This theoretical insight is empirically validated in Section C.6.

## 2.4 Asymptotic convergence in distribution of cyclical step-sizes

Note that for constant or cyclical learning rate schemes, Theorem 2.1 can only guarantee an MSE bound of order $O(1)$. This naturally motivates a deeper investigation into the convergence properties of these schedules. Specifically, an SGD sequence as defined in (1.1) with a constant step size $\eta_t \equiv \eta$

can be interpreted as an aperiodic Markov chain. Under standard regularity conditions, it is well-known that the iterates $\theta_t$ converge weakly to a stationary distribution. However, as discussed, recent empirical work has highlighted the benefits of periodic or cyclical step-size schedules [Loshchilov and Hutter, 2017, Smith, 2023, Wang et al., 2023], such as $\eta_t \equiv \eta_{t \bmod T}$. In this setting, the time-varying learning rate breaks the asymptotic stationarity of the SGD chain. Nonetheless, the periodic structure of the step-size schedule induces a corresponding periodicity in the asymptotic behavior of the iterates. Such non-stationary processes, characterized by recurring statistical properties over time, are known as cyclostationary processes, which we briefly introduce below.

**Definition 2.3** (Cyclostationary process). *A stochastic process $\{X_t\}_{t \in \mathbb{R}}$ is said to be cyclostationary with period $T > 0$ if it holds that for all $s \in [T]$, and $i \in \mathbb{N}$, $\{X_i, \ldots, X_{i+s}\} \overset{d}{=} \{X_{i+T}, \ldots, X_{i+s+T}\}$.*

Cyclostationary process were introduced as a model of communications systems in Bennett [1958] and Franks [1969], later finding wide use in econometrics [Parzen and Pagano, 1979] as well as atmospheric sciences [Bloomfield et al., 1994] – the reader is encouraged to look into [Gardner et al., 1994, Napolitano, 2016], and the references therein for an introduction and a comprehensive list of all its applications. In the context of SGD, it is instructive to look at the iterative random function construction of the cyclostationary process, as introduced by Bonnerjee et al. [2024]:

$$X_t = g(\phi_t, \mathcal{F}_t), \ \mathcal{F}_t = \sigma(\varepsilon_s : s \le t), \ \phi_t = \phi_{t \bmod T}, \ \text{for some period } T \in \mathbb{N}. \tag{2.3}$$

This representation suggests an immediate connection to the SGD iterates $\theta_t$ in (1.1), which, in the case of cyclical learning schedules, can be represented as

$$\theta_t = F_{\xi_t}(\eta_t, \theta_{t-1}), \ \eta_t = \eta_{t \bmod T}, \ \text{for some period } T \in \mathbb{N}, \ \text{with } F_\xi(\eta, \theta) = \theta - \eta \nabla f(\theta, \xi). \tag{2.4}$$

Equations (2.3) and (2.4) suggest an immediate connection between the cyclostationary process and SGD with cyclic learning rate, with the choice $\phi_t = \eta_t$ for $t \in [T]$. The following result, proved in appendix Section A.2, makes this connection precise by establishing a novel asymptotic convergence result.

**Theorem 2.4.** *Suppose that Assumptions 2.1 and 2.3 hold for some $p > 2$. Let $\rho_p(\gamma)^p := (1 + \gamma L_p)^p - p\gamma L_p - p\mu\gamma, \ \gamma \in \mathbb{R}$, where $\mu$ and $L_p$ are as in Assumptions 2.1 and 2.3 respectively. Consider a periodic step-size schedule with fixed period $T$. Then there exist $T$ stationary processes $\pi_1, \ldots, \pi_T$ such that for all $\boldsymbol{\eta} := (\eta_1, \ldots, \eta_T) \in \mathbb{R}^T$ satisfying*

$$\rho_p(\eta_1) \ldots \rho_p(\eta_T) < 1, \tag{2.5}$$

*it holds that*

$$\theta_{nT+i} \overset{w}{\Rightarrow} \pi_i \ \text{as } n \to \infty \ \text{for all } i \in [T]. \tag{2.6}$$

*Moreover, if $\boldsymbol{\eta}$ further satisfies*

$$\min_s \mathcal{J}_p(s) < 1, \ \text{with } \mathcal{J}_p(s) = \sum_{k=1}^{T} \prod_{j=1}^{k} \rho_p(\eta_{s+j})^p, \tag{2.7}$$

*where $\eta_j = \eta_{j \bmod T}$ for $j > T$, then there exists a cyclostationary process $\pi$, such that*

$$\theta_n \overset{w}{\Rightarrow} \pi \ \text{as } n \to \infty. \tag{2.8}$$

*Remark* 2.4. Note that (2.5) ensures that $\rho_p(\eta_{s^\star}) < 1$, where $s^\star = \arg\min_s \mathcal{J}_p(s)$. In contrast to the SGD with constant learning rate, none of the conditions (2.5) and (2.7) presupposes that $\eta_i$'s are required to satisfy $\rho_p(\eta_i) < 1$ for each $i \in [T]$. In particular, at least some of the $\eta_i$'s may be taken to be large, which helps in faster convergence, which is also seen empirically in Figure 4. This result underpins the flexibility and the resulting popularity of the periodic step-size schedule over its constant counter-part, guaranteeing convergence under very mild conditions (2.5) and (2.7) respectively.

# 3 Simulation

To validate our theoretical analysis, we conduct an empirical study of SGD with various learning rate schedules. The goal is to assess how different step-size strategies affect convergence behavior and

mean squared error, and how these compare with the theoretical predictions in Theorems 2.1, 2.2 and 2.4. For simplicity, all experiments use a simple linear regression model with known ground truth and are repeated across multiple Monte Carlo runs to estimate average performance and variability. In particular, Section 3.1 provides the model specifications. Section 3.2, we study the linearly decaying to zero (Linear-D2Z) schedule $\eta_t = \eta_0(1 - t/n)$, confirming its "best-of-both-worlds" performance—fast early convergence and diminishing final error—as predicted by Theorem 2.2. Finally, Section 3.3 examines cosine learning rate schedules of the form $\eta_t = \eta_0(1 + \cos(\pi t/T))$, where we observe periodic error fluctuations that empirically confirm the cyclostationary behavior predicted by Theorem 2.4. Additional empirical studies can be found in Appendix C, where we also include a particularly illuminating comparative study between the different schedules in Section C.6. All the code files are available in GitHub.

## 3.1 Model specification

All the experiments are based on the following simple linear regression model:

$$y_i = \theta^{(0)} + \theta^{(1)}x_i + \varepsilon_i, \quad \varepsilon_i \sim \mathcal{N}(0,1) \text{ i.i.d., } \theta^\star = (\theta^{(0)}, \theta^{(1)})^\top \in \mathbb{R}^2. \tag{3.1}$$

where $(x_i, y_i) \in \mathbb{R}^2$ denotes the observed data and $\theta^\star \in \mathbb{R}^2$ is the unknown parameter. The true parameter vector is fixed at $\theta^\star = (2, -3)^\top$ throughout all experiments. For all the subsequent simulation studies, we initialize the SGD chain at $(0,0)^\top$, which provides sufficient distance for meaningful comparisons across different learning rate schedules, while not being so far away from the ground truth so that it fails to converge and denies us the full picture. Subsequently, we focus on an empirical evaluation of both the convergence trajectory and the distribution of the final error across different learning rate strategies.

## 3.2 Linear-D2Z rate

We consider Linear-D2Z schedules of the form $\eta_t = \eta_0(1 - t/n)$, which, despite their widespread use, have received comparatively little theoretical attention. Similar to Section 3.3, we let $\eta_0 \in \{0.01, 0.05, 0.1\}$, and for each experiment, the mean errors are estimated via $n_{iter} = 500$ many independent repetitions. Firstly, to analyze the non-asymptotic MSE of the end-term SGD iterates, $\theta_n$, we run SGD on the same regression task with $n \in \{100, 200, \dots, 10^4\}$, using 500 independent repetitions for each $n$. Figure 1 displays the terminal squared error, which decays polynomially with $n$, in line with Theorems 2.1 and 2.2.

The appeal of the Linear-D2Z schedule lies in its hybrid structure: as explained in Remark 2.3, early iterations benefit from relatively large step sizes, enabling rapid descent—potentially faster than a constant-rate scheme. Later, the schedule tapers off, reducing variance and yielding low final error. We numerically investigate this as follows. For a fixed $n = 10^4$, Figure 2 shows the first 100 iterations for $\eta_0 \in \{0.05, 0.1, 0.5\}$. Across all settings, the error drops sharply, even under the high-variance $\eta_0 = 0.5$ case, demonstrating the robustness of the approach. Later in training, as shown in Figure 3) the error decay appears nearly linear before plateauing, with the stabilization occurring earlier for larger $\eta_0$.

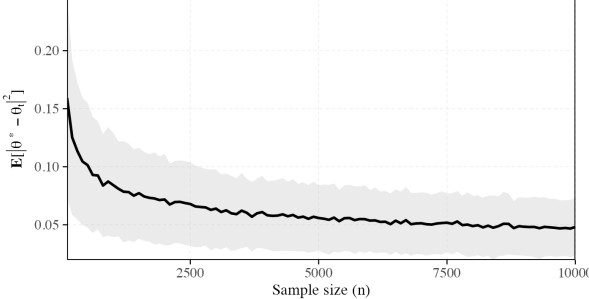

Figure 1: Plot of the terminal MSE estimate averaged over 500 SGD runs for $n = 100$ to $n = 10^4$.

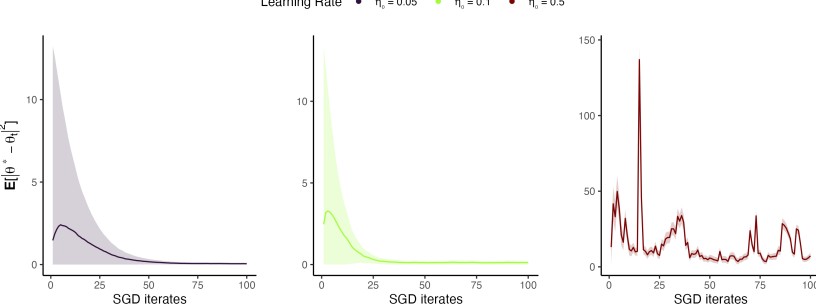

Figure 2: Plot of the MSE estimate averaged over 500 SGD runs for $n = 10^4$ iterations under a `Linear-D2Z` schedule, observing from step 1 to 100.

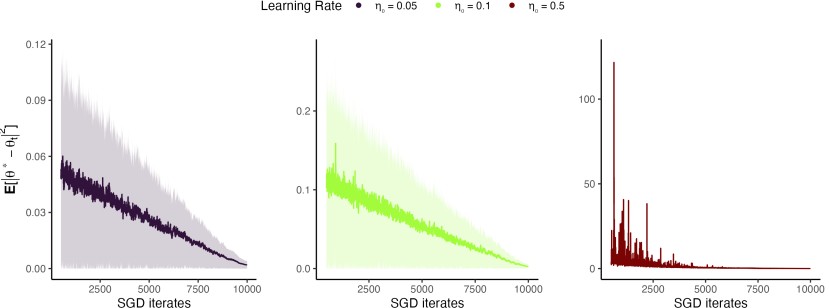

Figure 3: Plot of the MSE estimate averaged over 500 SGD runs for $n = 10^4$ iterations under a `Linear-D2Z` schedule, observing from step 500 onward.

### 3.3 Cosine learning rate

As an example of the cyclical learning rate, we employ the widely-used cosine scheduling. Specifically, we employ the schedule:

$$\eta_t = \eta_0 \left( 1 + \cos\left( \frac{2\pi t}{T} \right) \right), \tag{3.2}$$

where $\eta_0$ is the base learning rate and $T$ denotes the period.

To assess the behavior induced by such schedules, we perform online SGD for $n = 10^4$ iterations with $\eta_0 \in \{0.01, 0.05, 0.1\}$ and $T = 3$, averaging results over $n_{iter} = 500$ independent trials. Figure 4 presents the resulting mean squared error (MSE) trajectories. Across all settings, the MSE exhibits an exponentially fast decay from the initial points before exhibiting persistent fluctuations about a steady-state level. The periodicity, as predicted by Theorem 2.4 is not apparent from this plot. To further probe this structure, we examine the *standard deviation* of the MSE across runs over the final 100 iterations in Figure 5. Despite autocorrelation between estimates (due to small step sizes), periodicity remains visible in the standard deviation curves. Even for $\eta_0 = 0.01$, where the process converges slowly, the periodicity of order 3 is easily discernible on the plot. This empirically confirms Theorem 2.4: despite the huge auto-correlation between successive iterations, with cyclical learning rates SGD iterates $\theta_t$ do not settle but oscillate periodically, reflecting the learning rate's structure. The oscillation's amplitude and frequency depend on $\eta_0$, with larger values causing stronger fluctuations and faster initial progress. This nuanced behavior highlights the balance between exploration and convergence enabled specifically by periodic schedules. For comparison, Figure 6 shows standard deviation curves under constant learning rates, which lack periodicity, confirming that the patterns in Figure 5 arise from the cosine schedule.

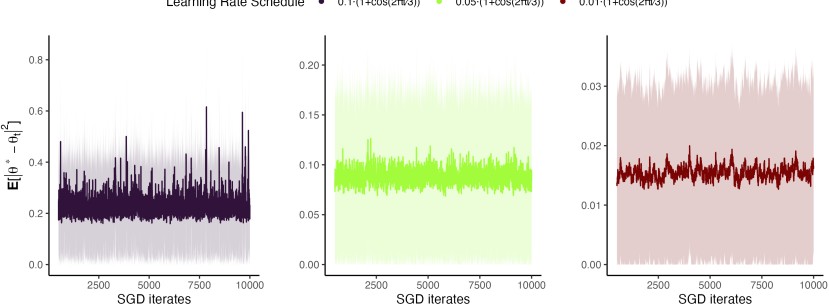

Figure 4: MSE estimates over 500 SGD runs ($10^4$ steps) with cosine learning rates, observing from step 500 onward. Periodic error fluctuations are evident, indicating cyclostationary dynamics.

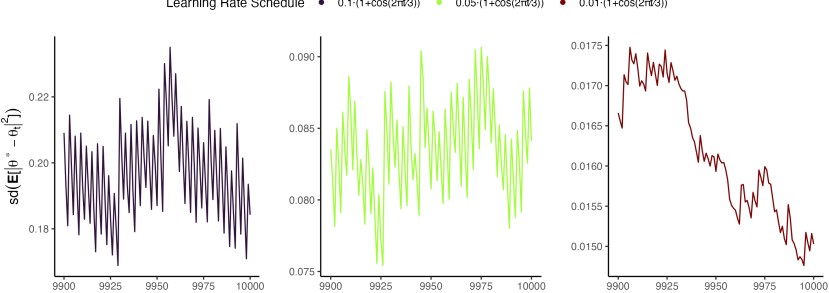

Figure 5: Standard deviation of the MSE over the final 100 iterations (across 500 runs) with cosine learning rates. Periodic fluctuation is clearly observed, even at a small $\eta_0$.

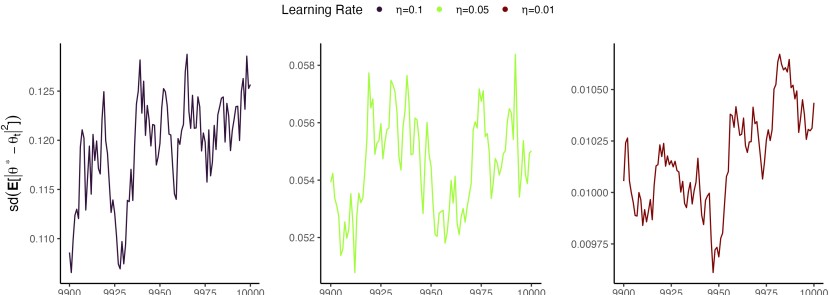

Figure 6: Standard deviation of the MSE over the final 100 iterations with constant learning rates. No periodicity is observed.

## 4 Conclusion

Sharp theoretical MSE bounds offer critical insights into the behavior of SGD for given learning rate schedules, yet most prior work has focused on polynomially decaying step sizes, often sacrificing convergence speed for statistical tractability. To the best of our knowledge, this paper is the first to systematically develop a unified framework that provides explicit MSE upper bounds for a broad class of learning rates. In particular, we establish novel convergence guarantees for cyclical and linearly decaying to zero (`Linear-D2Z`) learning rates—two popular but previously under-theorized choices—shedding light on their strong empirical performance. Our results motivate further exploration beyond the convex setting into non-convex and non-smooth landscapes, with an emphasis on understanding the statistical behavior of these schedules, including the potential for central limit theorems and refined uncertainty quantification. In this context, this work provides new insights into

the practically important yet theoretically underexplored area of learning rate selection, and serves as a foundation for bridging practical success and theoretical understanding of SGD across diverse learning schedules regimes.

## Acknowledgments and Disclosure of Funding

We sincerely thank the program chair, senior area chair, area chair, and the five reviewers for their constructive feedback and involved discussion, which has greatly improved the clarity of our paper. Jiaqi Li's research is partially supported by the NSF (Grant NSF/DMS-2515926). Wei Biao Wu's research is partially supported by the NSF (Grant NSF/DMS-2311249).

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

# A  Postponed Proofs

## A.1  Proof of Theorem 2.1

*Proof of Theorem 2.1.* By recursively applying Lemma B.2, we have

$$\|\theta_n - \theta^\star\|_p^2 \leq \prod_{k=1}^{n}(1 - c_0\eta_k)|\theta_0 - \theta^\star|^2 + 3(p-1)M_p^2 \sum_{j=1}^{n}\eta_j^2 \prod_{k=j+1}^{n}(1 - c_0\eta_k).$$

The proof is completed by noting that

$$\prod_{k=1}^{n}(1 - c_0\eta_k) \leq \exp\left\{ - c_0\sum_{k=1}^{n}\eta_k\right\}.$$

$\square$

## A.2  Proof of Theorem 2.4

*Proof of Theorem 2.4.* Suppose $F_{X,\gamma}(\theta) = \theta - \gamma\nabla f(\theta, \xi)$ encodes the iterative random function governing the SGD trajectory (1.1). Fix $i \in [T]$. For (2.6), observe that

$$\theta_{nT+i} = F^i_{(\xi_{(n-1)T+i+1},\ldots,\xi_{nT+i})}(\theta_{(n-1)T+i}, \boldsymbol{\eta}),$$

where, for $\mathbf{X} = (X_1, \ldots, X_T)$, we define $F^i_{\mathbf{X}}(\theta, \boldsymbol{\eta}) := F_{X_1,\eta_{i+1}} \circ \ldots \circ F_{X_T,\eta_{i+T}}(\theta)$, with $\eta_{i+s} = \eta_{i+s \bmod T}$ with slight abuse of notation. Applying Theorem 2.2 of Li et al. [2024a] successively on the function compositions of $F^i$, it holds that

$$\|F^i_{\mathbf{X}}(\theta, \boldsymbol{\eta}) - F^i_{\mathbf{X}}(\theta', \boldsymbol{\eta})\| \leq \rho_p(\eta_1)\ldots\rho_p(\eta_T),$$

from which, (2.6) follows in light of (2.5) and Theorem 2.2 of Li et al. [2024a]. To ensure (2.8), note that if $\pi$ is a cyclostationary process with period $T$ defined on $\mathbb{R}^d$, then $X \sim \pi$ iff $(X_1, \ldots, X_T) \sim \tilde{\pi}$ for some stationary process $\pi$ on $\mathbb{R}^{d \times T}$. Therefore, it is enough to show

$$(\theta_{nT+s} : \ldots : \theta_{(n+1)T+s}) \to \tilde{\pi} \tag{A.1}$$

for some stationary process $\tilde{\pi}$ on $\mathbb{R}^{d \times T}$, and some $s \in [T]$. Choose $s = s^\star$ such that

$$s^\star = \arg\min_{s \in [T]} \sum_{k=1}^{T}\prod_{j=1}^{k}\rho_p(\eta_{s+j})^p.$$

Define

$$\tilde{F}_{\mathbf{X}}(\boldsymbol{\Theta}, \boldsymbol{\eta}) = (F_{X_1,\eta_{s+1}}(\theta_t), F_{X_2,\eta_{s+2}} \circ F_{X_1,\eta_{s+1}}(\theta_t), \ldots, F_{X_T,\eta_{s+T}} \circ \cdots \circ F_{X_1,\eta_{s+1}}(\theta_t)),$$

where

$$\boldsymbol{\Theta} = (\theta_1, \ldots, \theta_T) \in \mathbb{R}^{d \times T}.$$

Clearly, one derives

$$\|\tilde{F}_{\mathbf{X}}(\boldsymbol{\Theta}, \boldsymbol{\eta}) - F^i_{\mathbf{X}}(\boldsymbol{\Theta}', \boldsymbol{\eta})\|_p^p = \sum_{k=1}^{T}\left\|F_{X_k,\eta_{s+k}} \circ \cdots F_{X_1,\eta_{s+1}}(\theta_t) - F_{X_k,\eta_k} \circ \cdots F_{X_1,\eta_1}(\theta'_t)\right\|_p^p$$

$$\leq \|\theta_t - \theta'_t\|_p^p \sum_{k=1}^{T}\prod_{j=1}^{k}\rho_p(\eta_{s+j})^p. \tag{A.2}$$

Writing (1.1) as

$$(\theta_{nT+s+1,\ldots,\theta_{(n+1)T+s}}) = \tilde{F}_{\xi_{nT+s+1},\ldots,\xi_{(n+1)T+s}}((\theta_{(n-1)T+s+1,\ldots,\theta_{nT+s}}), \boldsymbol{\eta}),\ n \geq 1$$

yet another application of Theorem 2.2 of Li et al. [2024a] yields (A.1) in light of (A.2) and (2.7). $\square$

## A.3 Proof of Corollary 2.2

*Proof of Corollary 2.2.* When $\eta_t = \eta(1 - t/n)$, the first term in (2.2) becomes $\exp\{-c_0\eta(n - 1)/2\}|\theta_n - \theta^\star|^2$. For the second term, set $m = n - j$. Then $m = 0, 1, \ldots, n - 1$, and

$$\eta_j = \eta\frac{n-j}{n} = \eta\frac{m}{n}, \qquad \sum_{k=j+1}^{n} \eta_k = \sum_{k=0}^{m-1} \eta\frac{k}{n} = \frac{\eta}{n}\frac{(m-1)m}{2} = \frac{\eta}{2n}(m^2 - m).$$

Let $S_n = \sum_{j=1}^{n} \eta_j^2 \exp\left\{ - c_0 \sum_{k=j+1}^{n} \eta_k\right\}$, then since $m \leq n$,

$$S_n = \sum_{m=0}^{n-1} \left(\eta\frac{m}{n}\right)^2 \exp\left\{-c_0 \frac{\eta}{2n}(m^2 - m)\right\} \leq \eta^2 \exp\left(\frac{c_0\eta}{2}\right) \sum_{m=0}^{n-1} \frac{m^2}{n^2} \exp\left\{-\frac{c_0\eta m^2}{2n}\right\}.$$

The sum can be further bounded by the integration. Since the function $x^2 \exp\{c_0\eta x^2/2\}$ is eventually decreasing with $x$, we have

$$\sum_{m=0}^{n-1} \frac{m^2}{n^2} \exp\left\{-\frac{c_0\eta m^2}{2n}\right\} \leq \frac{C'}{\sqrt{n}} \int_0^\infty x^2 \exp\left\{-\frac{c_0\eta x^2}{2}\right\} dx = O(1/\sqrt{n})$$

where $C'$ is a universal constant, and the inequality above is obtained by substituting $x = m^2/n$. This completes the proof. $\square$

## B Auxiliary Section

In this section we collect two crucial auxiliary results that contribute towards the proof of Theorem 2.1.

**Lemma B.1** (Rio's inequality [Rio, 2009]). *Let $X \in \mathbb{R}^d$ and $Y \in \mathbb{R}^d$ be two random vectors such that $\mathbb{E}|X|^p < \infty$ and $\mathbb{E}|Y|^p < \infty$ for some $p \geq 2$. Then we have*

$$\|X + Y\|_p^2 \leq \|X\|_p^2 + (p - 1)\|Y\|_p^2.$$

**Lemma B.2.** *Consider the SGD iterates $\{\theta_t\}_{t\geq 1}$ in (1.1). Suppose that Assumptions 2.1 and 2.3 for $p \geq 2$. Then, for some constant $c_0 > 0$ such that for all $t \geq 1$,*

$$c_0 \leq \min\left\{\frac{1}{\eta_t}, 2\mu - (6p - 5)L_p^2\eta_t\right\},$$

*we have, for all $t \geq 1$,*

$$\|\theta_t - \theta^\star\|_p^2 \leq (1 - c_0\eta_t)\|\theta_{n-1} - \theta^\star\|_p^2 + 3(p - 1)\eta_t^2 M_p^2.$$

### B.1 Proof of Lemma B.2

*Proof of Lemma B.2.* Since $\xi_t$, for $t \geq 1$, are i.i.d. random samples, it follows from the tower rule that

$$\mathbb{E}[\nabla f(\theta_{n-1}, \xi_n) - \nabla F(\theta_{n-1}) \mid \theta_{n-1}] = 0.$$

Therefore, by applying Rio's inequality in Lemma B.1, for $p \geq 2$, we have

$$\|\theta_n - \theta^\star\|_p^2 \leq \|\theta_{n-1} - \theta^\star - \eta_n\nabla F(\theta_{n-1})\|_p^2 + (p-1)\eta_n^2\|\nabla f(\theta_{n-1}, \xi_n) - \nabla F(\theta_{n-1})\|_p^2$$
$$=: \mathbb{I}_1 + \mathbb{I}_2.$$

We shall bound the two parts $\mathbb{I}_1$ and $\mathbb{I}_2$ separately. For the first part $\mathbb{I}_1$, note that $\nabla F(\theta^\star) = 0$ and by the triangle inequality, we have

$$\mathbb{I}_1 = \|\theta_{n-1} - \theta^\star - \eta_n\nabla F(\theta_{n-1})\|_p^2$$
$$= \Big\|\langle\theta_{n-1} - \theta^\star, \theta_{n-1} - \theta^\star\rangle - 2\eta_n\langle\theta_{n-1} - \theta^\star, \nabla F(\theta_{n-1}) - \nabla F(\theta^\star)\rangle$$
$$\quad + \eta_n^2\langle\nabla F(\theta_{n-1}) - \nabla F(\theta^\star), \nabla F(\theta_{n-1}) - \nabla F(\theta^\star)\rangle\Big\|_{p/2}$$
$$\leq \Big\|\langle\theta_{n-1} - \theta^\star, \theta_{n-1} - \theta^\star\rangle - 2\eta_n\langle\theta_{n-1} - \theta^\star, \nabla F(\theta_{n-1}) - \nabla F(\theta^\star)\rangle\Big\|_{p/2}$$
$$\quad + \eta_n^2\big\|\nabla F(\theta_{n-1}) - \nabla F(\theta^\star)\big\|_p^2.$$

By applying Assumption 2.1 to the first term and Assumption 2.3 to the second term, we can obtain

$$\mathbb{I}_1 \leq (1 - 2\eta_n\mu + \eta_n^2 L_p^2)\|\theta_{n-1} - \theta^\star\|_p^2.$$

Regarding the second part $\mathbb{I}_2$, since $\nabla F(\theta^\star) = 0$, we have

$$\begin{aligned}&\|\nabla f(\theta_{n-1}, \xi_n) - \nabla F(\theta_{n-1})\|_p\\&\leq \|\nabla f(\theta_{n-1}, \xi_n) - \nabla f(\theta^\star, \xi_n)\|_p + \|\nabla F(\theta_{n-1}) - \nabla F(\theta^\star)\|_p + \|\nabla f(\theta^\star, \xi_n)\|_p.\end{aligned}$$

Hence, by Assumption 2.3, we can achieve

$$\|\nabla f(\theta_{n-1}, \xi_n) - \nabla F(\theta_{n-1})\|_p^2 \leq 6L_p^2\|\theta_{n-1} - \theta^\star\|_p^2 + 3\|\nabla f(\theta^\star, \xi_n)\|_p^2.$$

Combining results from $\mathbb{I}_1$ and $\mathbb{I}_2$, we can obtain

$$\|\theta_n - \theta^\star\|_p^2 \leq (1 - 2\eta_n\mu + (6p-5)\eta_n^2 L_p^2)\|\theta_{n-1} - \theta^\star\|_p^2 + 3(p-1)\eta_n^2\|\nabla f(\theta^\star, \xi_n)\|_p^2.$$

This can directly lead to the desired inequality. $\qquad\square$

## C  Additional simulations

In this section, we collect some additional simulation studies, complimenting the numerical exercises of Section 3. In particular, in Appendix C.1, we begin with constant learning rates ($\eta_t \equiv \eta$), which serve as a baseline and exhibit the expected bias-variance tradeoff, with final error scaling linearly in $\eta$. Appendix C.2 turns to polynomially decaying learning rates, $\eta_t = \eta_0 t^{-\beta}$, and demonstrates how different values of $\beta \in (1/2, 1)$ influence the tradeoff between fast initial descent and long-run variance control, consistent with the structure of Theorem 2.1. Moreover, Appendix C.3 examines alternating schedules that interleave two polynomial decay rates or combine a constant rate with a decaying one, highlighting how the more aggressive schedule tends to dominate long-run behavior. Finally, Appendix C.6 includes a detailed comparison study of the different step-sizes considered in this paper, with particular focus into the initial phase, as well as asymptotic behavior upon convergence.

### C.1  Constant learning rate

To ground our analysis, we start with the familiar case of constant learning rates—long favored for their simplicity, but known to encode a fundamental tradeoff. We test three fixed values, $\eta = 0.1, 0.05$, and $0.01$, and track their performance over $10^4$ SGD iterations. The patterns are predictable but instructive: larger step sizes yield faster initial progress, yet settle into higher-variance regimes; smaller ones move more cautiously, but converge closer to the optimum with lower final error.

Because the early dynamics often involve rapid error reduction—sometimes several orders of magnitude—we focus the MSE plot on later stages: starting from $t = 100$ for $\eta = 0.1$ and $0.05$, and from $t = 500$ for $\eta = 0.01$. This lets us zoom in on the asymptotic behavior, where the long-term effects of each learning rate become more clearly visible.

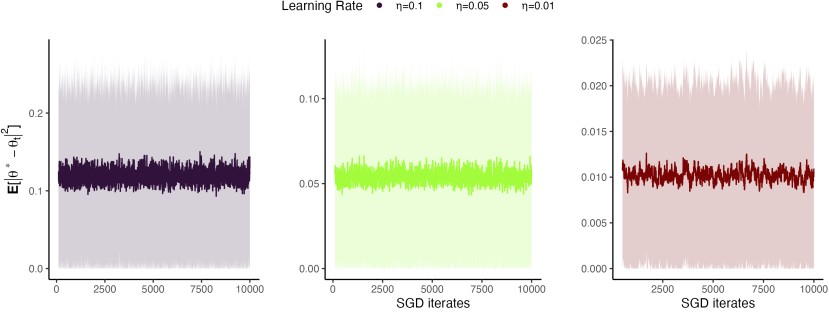

Figure 7: MSE estimates over 500 SGD runs ($10^4$ steps) with constant learning rates.

The empirical trajectories in Figure 7 reflect this tradeoff. Larger learning rates lead to faster early reduction in error but exhibit higher variance and stabilize farther from the optimum. In contrast, smaller learning rates result in slower progress but achieve significantly lower terminal error.

To complement this, Figure 8 plots the final mean squared error against a dense grid of fixed learning rates ranging from $\eta = 0.01$ to $\eta = 0.1$. The trend is unmistakable: terminal MSE scales linearly with $\eta$, matching the $O(\eta)$ variance bound predicted by theory. This reinforces the fundamental tension in fixed-rate SGD: speed comes at the cost of noise, and there's no single value of $\eta$ that avoids the tradeoff entirely.

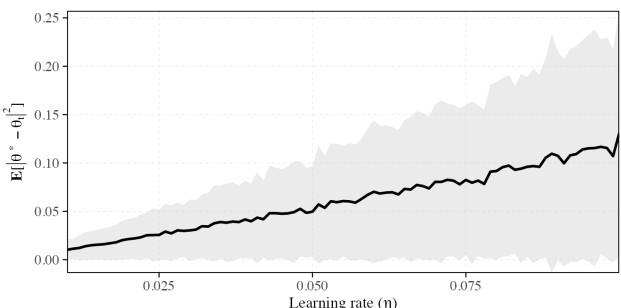

Figure 8: Plot of the terminal MSE estimate averaged over 500 SGD runs for $10^4$ steps, for constant values for $\eta$ between 0.01 and 0.1.

## C.2 Polynomially decaying learning rate

We now turn to the classical regime of polynomially decaying learning rates. These schedules take the form $\eta_t = \eta_0 t^{-\beta}$, where $\eta_0 > 0$ and $\beta \in (0, 1]$ controls the rate of decay. The general non-asymptotic error bound given in Theorem 2.1 applies to this setting directly, and allows us to capture the tradeoffs these schedules induce. In particular, when $\beta > \frac{1}{2}$, the sum $\sum_{t=1}^{\infty} \eta_t^2$ converges, ensuring that variance contributions decay to zero; whereas the condition $\beta < 1$ guarantees $\sum_{t=1}^{\infty} \eta_t = \infty$, which is necessary for the bias to vanish. These facts jointly imply that SGD with $\beta \in (\frac{1}{2}, 1)$ is consistent and convergent, with error rates depending sensitively on the balance between the two terms in the bound.

To explore these effects empirically, we simulate SGD with learning rates of the form $\eta_t = \eta_0 t^{-\beta}$ using two values of $\beta$: 0.505 and 0.75, each tested with base rates $\eta_0 = 0.1, 0.05, 0.01$. Figure 9 shows the mean squared error over time for these settings, averaged over 500 independent runs. The results highlight the central tradeoff: smaller values of $\beta$ yield faster initial descent but larger long-run fluctuations, while larger $\beta$ dampen early progress but reduce terminal error. This qualitative pattern matches the structure of Theorem 2.1, in which the exponential forgetting term dominates early on, and the variance accumulation term becomes decisive in the long run.

Because the early dynamics often involve rapid error reduction—sometimes several orders of magnitude—we focus the MSE plot on later stages: starting from $t = 1000$ for $\eta = 0.1, \beta = 0.505$, $t = 4000$ for $\eta = 0.05, \beta = 0.505$ and $t = 5000$ for $\beta = 0.75$. This lets us zoom in on the asymptotic behavior, where the long-term effects of each learning rate become more clearly visible.

An interesting feature of these experiments is the relative importance of $\beta$ compared to $\eta_0$. While smaller base rates do modestly influence early error and variance, the dominant effect stems from the decay exponent. The case $\beta = 0.505$, being just above the variance threshold, achieves a strong balance between speed and consistency—converging faster than $\beta = 0.75$ while eventually achieving comparably low error. This behavior reflects the non-asymptotic structure predicted by Theorem 2.1, which separates the error into two components: a bias decay term, of the form $\exp(-c \sum_{t=1}^{n} \eta_t)$, and a cumulative variance term, of the form $\sum_{j=1}^{n} \eta_j^2 \exp(-c \sum_{t=j+1}^{n} \eta_t)$. For polynomial learning rates, the bias term vanishes polynomially in $n$, and the variance term converges to zero if $\beta > \frac{1}{2}$, but only slowly. These dynamics explain the empirical behavior observed: $\beta = 0.505$ gives faster early convergence due to slower bias decay, while $\beta = 0.75$ leads to more effective long-run averaging, with reduced variance. The simulations thus concretely illustrate the tension between forgetting and fluctuation that the theory encodes, and validate the asymptotics of polynomial decay schedules as captured by Theorem 2.1.

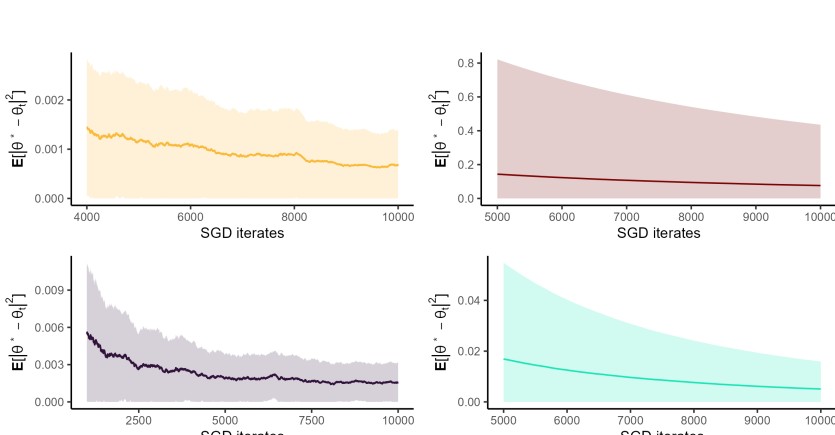

Figure 9: MSE estimates over $500$ SGD runs ($10^4$ steps) with polynomially decaying learning rates.

### C.3 Alternating polynomially decaying learning rate

Building on our observations from the previous experiments, we explore combining the strengths of both polynomially decaying and cyclical approaches. In particular, the cyclical schedules demonstrate superior initial convergence through their ability to take larger steps early in optimization, while polynomial decay provides better asymptotic properties and lower final error. To this end, we tried two forms of dual schedules:

1. Set a base rate $\eta_0$ and alternate between two polynomial decay rates or a polynomial decay rate and a constant rate. Using two values for $\eta_0$ $(0.1, 0.05)$, a constant rate and two exponents for polynomial decay rates $\beta = 0.505, 0.75$ gives us six total combinations. In all of these cases, we present the plots for the mixture of a decay schedule and a constant rate from $t = 500$ and for the mixture of two polynomial decay schedules from $t = 1000$, again to observe the general trend rather than the initial decay by several orders of magnitude in a short amount of time.

2. Set a fixed polynomial decay rate and alternate between two base rates, a "large" one and a "small" one. Plotted from $t = 2000$ for the same consideration as above.

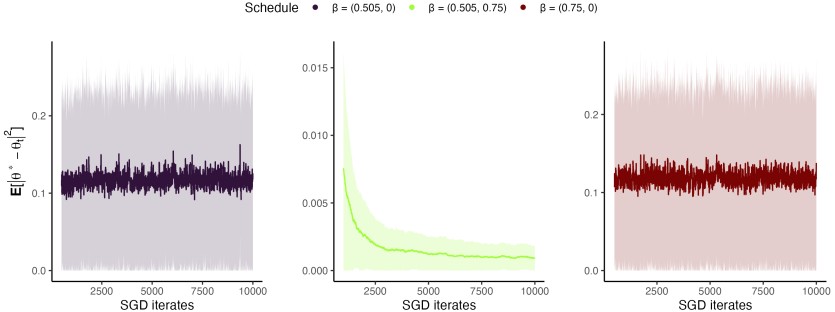

Figure 10: Plot of the MSE estimate averaged over $500$ SGD runs for $10^4$ steps.

For both $\eta_0 = 0.1$ and $\eta_0 = 0.05$, if the constant learning rate schedule is one of the two included, the process seems to behave in the same way over the long run. However, when we alternate between $\beta = 0.505$ and $\beta = 0.75$, we get an outcome more similar to just selecting $\beta = 0.505$. In all of these cases, the larger learning rate plays the dominant role in determining the convergence rate and terminal error of the process.

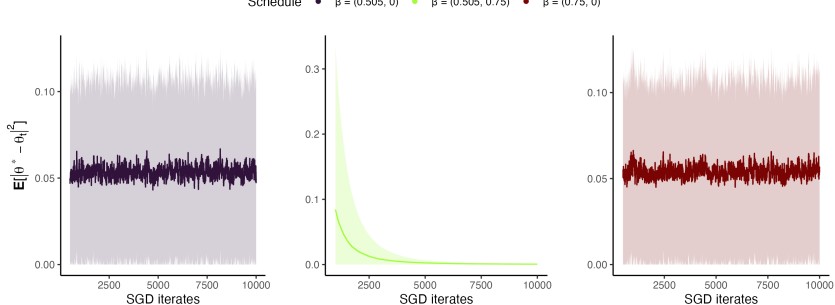

Figure 11: Plot of the MSE estimate averaged over 500 SGD runs for $10^4$ steps.

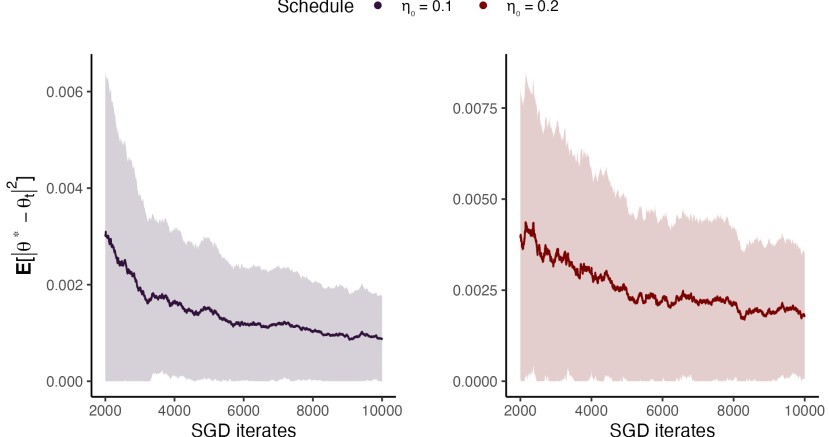

Figure 12: Plot of the MSE estimate averaged over 500 SGD runs for $10^4$ steps.

The two plots look almost identical and even have error rates that seem to scale sublinearly with $\eta_0$. Compared with the earlier plots in this section and our observations in section 3.3, we have empirical evidence that in polynomial decay and mixed polynomial decay regimes, the choice of the rate of decay $\beta$ is far more important than the choice of the base rate $\eta_0$.

## C.4   Linear regression with randomly initialized coefficients

In addition to the experiments outlined above, we present the results of another simulation, with $n = 10^5$, $\theta^* = (0,0)$ and $\theta_0$ initialized randomly. The other settings remain as they were in the paper:

$$y_i = \theta^{(0)} + \theta^{(1)}x_i + \varepsilon_i, \quad \varepsilon_i \sim \mathcal{N}(0,1) \text{ i.i.d.}, \quad \theta^* = (\theta^{(0)}, \theta^{(1)})^\top \in \mathbb{R}^2,$$

where $(x_i, y_i) \in \mathbb{R}^2$ denotes the observed data and $\theta^* \in \mathbb{R}^2$ is the unknown parameter. Results are averaged over 500 SGD runs. We summarize results from iterations $500 - 5000$ within each SGD run in the table below. Mean SE SD in the table is taken over the last 100 iterations.

Table 1: Final and Minimum MSE for Cosine Learning Rate Schedules on Simulation Data

| Schedule | Final MSE | Min MSE | Final SE SD | Mean SE SD |
|---|---|---|---|---|
| $0.1 \cdot (1 + \cos(2\pi t/3))$ | 0.1713 | 0.1103 | 0.1743 | 0.2334 |
| $0.05 \cdot (1 + \cos(2\pi t/3))$ | 0.0805 | 0.0484 | 0.0777 | 0.0912 |
| $0.01 \cdot (1 + \cos(2\pi t/3))$ | 0.0158 | 0.0101 | 0.0136 | 0.0144 |

## C.5 Testing on MNIST

To demonstrate the validity of our empirical evaluation beyond elementary linear regression cases, we conducted additional experiments on the MNIST dataset using a high-dimensional classification task. Specifically, we trained a multiclass logistic regression model via stochastic gradient descent (SGD) under both the cosine and `Linear-D2Z` learning rate schedules. The goal of this evaluation is to assess whether our theoretical insights carry over to practical settings involving real-world, high-dimensional data. Each MNIST image is flattened into a vector $x \in \mathbb{R}^{784}$ and paired with a one-hot encoded label $y \in \{0, 1\}^{10}$. Given this input-target pair, we minimize the sigmoid loss $\mathcal{L}(x, y; \theta)$, where $\theta \in \mathbb{R}^{784 \times 10}$ denotes the model parameters. This setup is equivalent to minimizing a sum of binary logistic regression losses across classes in a one-vs-rest fashion. For the cosine schedule $\eta_t = \eta_0(1 + \cos(2\pi t/3))$, we ran SGD for $n = 5000$ iterations, anticipating convergence to a cyclostationary distribution. The outcome was indeed cyclostationary in nature, similarly to what was observed in Figure 4.

For the `Linear-D2Z` schedule $\eta_t = \eta_0(1 - t/n)$, we ran SGD in increments of 500 iterations, from $n = 500$ to $n = 5000$. Performance was evaluated in terms of both the average sigmoid loss and the classification accuracy (i.e., the proportion of correctly classified digits under $\arg\max_j \theta^\top x$). Results are presented in the table below.

Table 2: Sigmoid loss Estimate for the `Linear-D2Z` schedule on the MNIST dataset

| Number of Iterations | MSE | Standard Deviation |
|---:|:---:|---:|
| 500 | 0.0059 | 0.0046 |
| 1000 | 0.0042 | 0.0020 |
| 1500 | 0.0041 | 0.0015 |
| 2000 | 0.0034 | 0.0018 |
| 2500 | 0.0034 | 0.0013 |
| 3000 | 0.0035 | 0.0012 |
| 3500 | 0.0033 | 0.0011 |
| 4000 | 0.0032 | 0.0011 |
| 4500 | 0.0032 | 0.0012 |
| 5000 | 0.0031 | 0.0011 |

## C.6 Comparison of different learning rates

In this section, we numerically investigate the comparative performance of the different learning rate schedules analyzed in this paper. Specifically, we plot the estimates of $\mathbb{E}[|\theta^* - \theta_n|^2]$ against $n$ for $n = 10^4$, and five different learning rate schedules corresponding to Sections 2.3-3.3, along with the additional studies in the appendix C. For careful comparison of both the effect of initialization as well as behavior at convergence, we investigate the MSE of $\theta_n$ for the initial 100 iterates, in Figure 13, as well as the final 100 iterates, in Figure 14.

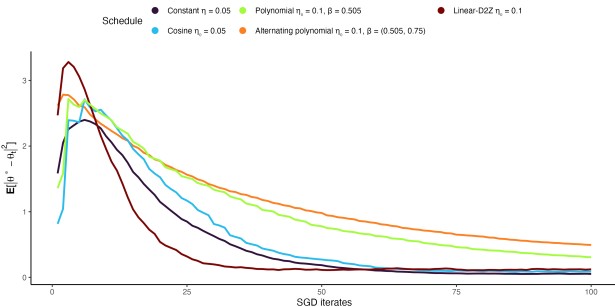

Figure 13: Plot of the evolution of the MSE estimate for the first 100 SGD iterations for five learning rate schedules.

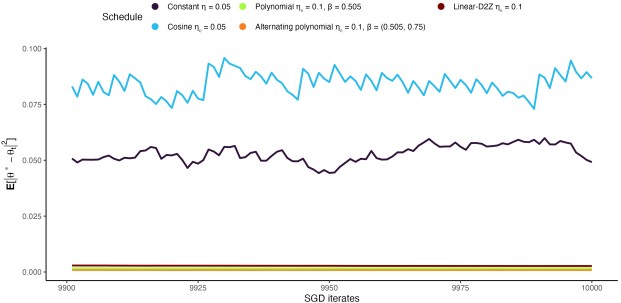

Figure 14: Plot of the evolution of the MSE estimate for the last 100 SGD iterations for five learning rate schedules.

In the early stages of the process, the models that have a non-decaying rate (constant learning rate and cosine schedule), as well as the `Linear-D2Z` model, which as discussed in Sections 2.3-3.3, exhibits superlinear convergence speed away from initialization, while the polynomial decay and mixed polynomial decay model both move more slowly. This is almost reversed in the later stages of the process - while all five models have converged to a small error, `Linear-D2Z` and the two polynomial decay-based models end up with a far smaller error estimate compared to the constant learning rate and the cosine schedule, which demonstrate a consistent bias. This demonstrates well-known theoretical results - larger learning rates converge faster but at the cost of a larger terminal bias. `Linear-D2Z` manages to reach an acceptably small error rather quickly and also enjoys a small terminal error due to being defined in a way that starts off with large step sizes that then taper off quickly, reinforcing its view as a "best-of-both-worlds" learning schedule.

