# OpenReview forum: "Asymptotic theory of SGD with a general learning-rate"
_NeurIPS.cc/2025/Conference — NeurIPS 2025 poster_

### Official Review · Reviewer_RA5h · 2025-07-01

**Clarity:** 3
**Significance:** 3
**Originality:** 3
**Rating:** 4
**Confidence:** 4

**Summary:**

This paper studied the convergence analysis of SGD with general learning rate that is more aligned with practice. They introduce a unified framework that quantifies the uncertainty of online SGD under arbitrary learning-rate choices. Specifically, they explored cyclical learning rates and linear decaying learning rate. Experiments validate the theoretical findings.

**Questions:**

1. Reference error: Line 271 ??
2. See weakness.

**Ethical Concerns:**

["NO or VERY MINOR ethics concerns only"]

**Final Justification:**

The author's response solves all of my concerns so that I will keep my score.

**Limitations:**

See weakness.

**Quality:**

3

**Strengths And Weaknesses:**

Strength: 1) the problem they studied is important to analyze the theoretical performance of practical working step sizes. 2) the analyzing tools are innovative. The theoretical finding is novel and inspiring.

Weaknesses: (1) For the cosine learning rate, which is widely used in practice, only experimental results are provided, with no accompanying theoretical guarantees.

(2) Theorems include too many assumptions, and their roles—as well as the classes of problems that satisfy them—are not clearly explained. For example, how can condition (2.7) be satisfied in practice?

(3) The experimental evaluation is limited. Although the authors validate various theoretical results through experiments, all experiments are conducted on linear regression problems. No large-scale experiments are included to demonstrate the scalability of the theoretical conclusions to more complex models.

(4) It would be helpful to include a direct comparison of convergence rates with those from existing literature to highlight the effectiveness and novelty of the new proof technique.

---

> ### Author Rebuttal · Authors · 2025-07-31
>
> We sincerely thank the reviewer for highlighting our practice-oriented analysis, our unified framework for quantifying uncertainty under arbitrary learning rate choices and our focus on cyclical and linearly decaying rates. We also appreciate the recognition of our innovative tools for analysis and the novel theoretical findings. In the following, we address your concerns one by one. We warmly welcome any other additional suggestions that you might have. If our responses seem adequate, we humbly ask you to consider an increase in your score.
>
> ## Weaknesses:
>
> - **For the cosine learning rate, which is widely used in practice, only experimental results are provided, with no accompanying theoretical guarantees.**
>
>   *Response.* We respectfully point out that the cosine learning rate schedule is a special case of the periodic learning rate schedule, discussed in Theorem 2.4.
>
> - **Theorems include too many assumptions, and their roles—as well as the classes of problems that satisfy them—are not clearly explained. For example, how can condition (2.7) be satisfied in practice?**
>
>   *Response.* We acknowledge the reviewer’s point that our assumptions, including Condition (2.7), could benefit from clearer contextualization and justification regarding practical applicability. In particular, (2.7) is a rather technical condition, and formal verification of such conditions requires knowledge of problem-specific parameters $p$ and $\mu$, which are often unknown. A detailed theoretical analysis might involve characterizing cyclical edge of stability, which promises to be an interesting research direction. In practice, however, as long as at least some of the $\eta_i$ are small, we expect this condition to hold—this is also evidenced by our numerical experiments. We have added this comment in a remark accompanying Theorem 2.4.
>
>   Similarly, equation (2.1) can be interpreted as requiring that step sizes are not too large. The rest of the technical assumptions, such as Assumptions 2.1 and 2.3, are standard in the literature (see, e.g., Polyak and Juditsky, *Acceleration of Stochastic Approximation by Averaging*, SIAM Optimization, 1992), and therefore we do not dwell on them in detail.
>
> - **The experimental evaluation is limited. All experiments are conducted on linear regression problems. No large-scale experiments are included.**
>
>   *Response.* To address this, we present results from a simulation using $n = 10^5$ and randomly initialized $\theta_0$:
>
>   $$
>   y_i = \theta^{(0)} + \theta^{(1)} x_i + \varepsilon_i,\quad \varepsilon_i \sim \mathcal{N}(0,1),\quad \theta^* = (\theta^{(0)}, \theta^{(1)})^\top \in \mathbb{R}^2
>   $$
>
>   where $(x_i, y_i) \in \mathbb{R}^2$ is the observed data. Results are averaged over 500 SGD runs.
>
>   **Table: Final and Minimum MSE for Cosine Learning Rate Schedules**
>
>   | Schedule                         | Final MSE | Min MSE | Final SE SD | Mean SE SD (last 100) | Iteration Range |
>   |----------------------------------|-----------|---------|--------------|------------------------|------------------|
>   | 0.1 · (1 + cos(2$\pi$t / 3))         | 0.1713    | 0.1103  | 0.1743       | 0.2334                 | 500–5000         |
>   | 0.05 · (1 + cos(2$\pi$t / 3))        | 0.0805    | 0.0484  | 0.0777       | 0.0912                 | 500–5000         |
>   | 0.01 · (1 + cos(2$\pi$t / 3))        | 0.0158    | 0.0101  | 0.0136       | 0.0144                 | 500–5000         |
>
>   We also conducted new experiments on MNIST. We trained a multiclass logistic regression model via SGD using both cosine and `Linear-D2Z` learning rate schedules. Images were flattened to $x \in \mathbb{R}^{784}$ with one-hot label $y \in \{0,1\}^{10}$, minimizing sigmoid loss $\mathcal{L}(x, y; \theta)$ with $\theta \in \mathbb{R}^{784 \times 10}$.
>
>   **Table: Sigmoid Loss Estimate for the `Linear-D2Z` Schedule on MNIST**
>
>   | Number of Iterations | MSE    | Standard Deviation |
>   |----------------------|--------|---------------------|
>   | 500                  | 0.0059 | 0.0046              |
>   | 1000                 | 0.0042 | 0.0020              |
>   | 1500                 | 0.0041 | 0.0015              |
>   | 2000                 | 0.0034 | 0.0018              |
>   | 2500                 | 0.0034 | 0.0013              |
>   | 3000                 | 0.0035 | 0.0012              |
>   | 3500                 | 0.0033 | 0.0011              |
>   | 4000                 | 0.0032 | 0.0011              |
>   | 4500                 | 0.0032 | 0.0012              |
>   | 5000                 | 0.0031 | 0.0011              |
>
> - **It would be helpful to include a direct comparison of convergence rates with those from existing literature.**
>
>   *Response.* We highlight that our work is among the first to derive convergence rates for linear-decaying and cyclical learning rates. For $p=2$, Theorem 2.2 aligns with Defazio et al. (2023), Corollary 2. Our rate of $1/\sqrt{n}$ matches theirs under strong convexity. In contrast to constant learning rates (which retain fixed bias) or polynomial decay (with slower $\exp(-cn^{1-\beta})$ convergence), linear decay achieves $\exp(-cn)$ forgetting. This showcases the novelty and strength of our results.
>
> ## Questions:
>
> - **Reference error: Line 271 ??**
>
>   *Response.* Thank you for pointing this out. The broken reference was for Figure 1 and has been corrected.

---

> > ### Comment · Reviewer_RA5h · 2025-08-04
> >
> > I thank the authors for the detailed response which solves my concerns. I will keep my score.

---

> > > ### Author Response · Authors · 2025-08-06
> > >
> > > Thank you very much for acknowledging our rebuttal. Given that your original concerns have now been resolved, we hope you might consider whether this further clarifies the contribution and impact of our paper sufficiently to justify a slightly higher score. We would greatly appreciate any additional feedback or reconsideration you might provide.

---

### Official Review · Reviewer_H11d · 2025-07-02

**Clarity:** 3
**Significance:** 3
**Originality:** 3
**Rating:** 4
**Confidence:** 3

**Summary:**

The authors propose a unified finite-sample analysis of vanilla SGD under arbitrary learning-rate schedules. 1) General moment bound (Thm 2.1). For any sequence $\{\eta_t\}$, even when $\eta_t$ depends on the optimisation horizon $n$, they bound the $p$-th moment error by an explicit two-term expression that separates the effect of the initialisation from a weighted sum of squared step-sizes. 2) Specialising Thm 2.1 yields what appears to be the first non-asymptotic MSE guarantee for the popular “linear-D2Z” policy $\eta_t=\eta(1-t/n)$ . 3) The authors prove convergence in distribution of the iterates to a cyclostationary process when $\eta_t$ is $T$-periodic, filling a gap in existing theory . 4) Simulations on a handful of convex models illustrate the contrasting transient and terminal errors of five schedules (constant, cosine, linear-D2Z, polynomial, mixed-polynomial).

**Questions:**

1) In the first line of Assumption 2.2, an identity matrix in the right-hand side of inequality is missing. Why is this assumption called local strong convexity? What is the point of this assumption brought in the main body when it's not used?
2) What is $|\cdot|_{p}$ in Equation (2.2)? Where is it defined?
3) Typo in line 200, "Section 2.3".
4) Please provide a citation for line 207.

**Ethical Concerns:**

["NO or VERY MINOR ethics concerns only"]

**Final Justification:**

The author’s reply fully resolves my concerns; I’m keeping my current rating.

**Limitations:**

Yes.

**Paper Formatting Concerns:**

None.

**Quality:**

3

**Strengths And Weaknesses:**

Strength:

1. Treating horizon-dependent and cyclical rates within the same inequality is elegant and may simplify future analyses.
2. The general bound recovers known optimal rates for polynomially decaying steps and constant steps (up to logarithmic factors).
3. Providing the first rigorous account of linear-decay and cyclical learning rates bridges a real gap between theory and what deep-learning practitioners actually use.
5. Despite heavy notation, the paper is well structured (Section 1 to  2 to 3) and key ideas are summarised before each proof .

Weaknesses:
1) In my view, it's good to bring the technical novelty in the proofs to the audience's attention.
2) The global strong convexity rules out the non-convex settings (deep nets, GANs) where exotic schedules are most used. So it could be nice to have these results under milder assumptions like PL inequality.

---

> ### Author Rebuttal · Authors · 2025-07-31
>
> We sincerely thank the reviewer for their thoughtful theoretically-oriented comments, highlighting our contributions on $p$th moment error bounds, our non-asymptotic MSE guarantee for the `Linear-D2Z` schedule, convergence to a cyclostationary process when the learning rate is periodic, and our comparison of multiple convex models pertaining to the theory discussed here. In the following, we address your concerns one by one. We warmly welcome any other additional suggestions that you might have. If our responses are satisfactory, we would greatly appreciate if you can consider a higher score.
>
> ## Weaknesses:
>
> - **In my view, it's good to bring the technical novelty in the proofs to the audience's attention.**
>
>   *Response.* Even though most of our proofs are generalizations of standard arguments, the ones for the cyclical step sizes required us to introduce novel tools to study the cyclo-stationarity of the sequence. To address this comment, we have added remarks to informally introduce the main ingredients of the proofs whenever appropriate.
>
> - **The global strong convexity rules out the non-convex settings (deep nets, GANs) where exotic schedules are most used. So it could be nice to have these results under milder assumptions like PL inequality.**
>
>   *Response.* We thank the reviewer for raising this pertinent question. Indeed, global strong convexity rules out the non-convex settings, but we argue that most of our proofs can be modified to also hold under the *local strong convexity* assumption when the number of local minima is finite; also see Zhong et al., *Online Bootstrap Inference with Non-Convex Stochastic Gradient Descent Estimator*, Preprint 2023. Unfortunately, for non-convex regimes with uncountably many local minima, the error bounds may not be obtained in an $\mathcal{L}_2$ sense, and it might be more convenient to look at regret bounds or convergence of $\nabla F(\theta_t)$.
>
>   Regarding the PL inequality, it is indeed a milder assumption, but its structure inherently prevents inference of the quantity $\\|\theta_n - \theta^\star\\|$; instead, PL inequality is more suited for bounding $|F(\theta_n) - F(\theta^\star)|$, which is not the focus of this paper. However, we agree that corresponding results may be obtained under PL inequality using not too dissimilar techniques, and we again thank the reviewer for pointing to this interesting research direction.
>
> ## Questions:
>
> - **In the first line of Assumption 2.2, an identity matrix in the right-hand side of inequality is missing. Why is this assumption called local strong convexity? What is the point of this assumption brought in the main body when it's not used?**
>
>   *Response.* This is a typo on our part; it should be "local strong concordance", and we are sorry for the confusion caused. Many examples, such as logistic regression, do not satisfy strong convexity but satisfy the local strong concordance property. The reason we mentioned this assumption was to highlight that our proofs also go through almost verbatim under this condition. We elaborate on this below.
>
>   A recurring theme of our proofs is to show that:
>
>   $$
>   |\theta - \theta^\star - \eta \nabla F(\theta)|^2 \leq (1 - \eta c)|\theta - \theta^\star|^2 \quad \text{for some } c > 0, \theta \in \mathbb{R}^d.
>   $$
>
>   ### Proof via strong convexity
>
>   $$
>   |\theta - \theta^\star - \eta \nabla F(\theta)|^2 = |\theta|^2 - 2 \eta (\theta - \theta^\star)^\top \nabla F(\theta) + \eta^2 |\nabla F(\theta)|^2 \leq (1 - 2\eta \mu + \eta^2 L^2) |\theta - \theta^\star|^2
>   $$
>
>   so the inequality follows for small enough $\eta$. Since we use decaying step size $\eta_t \propto t^{-\beta}$, it holds that $1 - 2\eta_t \mu + \eta_t^2 L^2 \leq 1 - \eta_t c$ for some $c > 0$ and large enough $t$.
>
>   ### Proof via local strong concordance
>
>   Define $\phi(u) = F(\theta^\star + u(\theta - \theta^\star))$ for $u \in [0, 1]$. Assume $|\theta-\theta^\star|\leq R$. Then:
>
>   $$
>   \phi''(0) \geq \mu^\star |\theta - \theta^\star|^2, \quad \phi''(u) \geq \phi''(0) \exp(-C|\theta - \theta^\star|u)
>   $$
>
>   and:
>
>   $$
>   (\theta - \theta^\star)^\top \nabla F(\theta) = \phi'(1) - \phi'(0) \geq \mu^\star |\theta - \theta^\star|^2 \cdot \frac{1 - \exp(-C|\theta - \theta^\star|)}{C|\theta - \theta^\star|} \geq \mu^\star C \exp(-R) |\theta - \theta^\star|^2
>   $$
>
>   which plugs into the earlier inequality to yield the desired bound. This explanation has also been added to the manuscript.
>
> - **What is $\\|\cdot\\|_p$ in Equation (2.2)? Where is it defined?**
>
>   *Response.* Assuming that you meant $\\|\cdot\\|_p$, we define it as $\\|X\\|_p = \left(\mathbb{E}[|X|^p]\right)^{1/p}$. This definition has now been added to the manuscript. We also clarify that $|\cdot|_p$ in your question refers to the vector $\ell_p$-norm, which is now also included in the text.
>
> - **Typo in line 200, "Section 2.3".**
>
>   *Response.* Thank you for pointing this out. This should have been "Section C.4", and has been corrected.
>
> - **Please provide a citation for line 207.**
>
>   *Response.* On line 207, we refer back to citations already introduced in lines 103–104: Loschilov and Hutter (*SGDR: Stochastic Gradient Descent with Warm Restarts*, ICLR 2017), Smith (*General Cyclical Training of Neural Networks*, 2023), and Wang et al. (*An Empirical Study of Cyclical Learning Rate on Neural Machine Translation*, 2023). We’ve added these citations to the relevant sentence for clarity.

---

### Official Review · Reviewer_8TRU · 2025-07-03

**Clarity:** 3
**Significance:** 3
**Originality:** 3
**Rating:** 5
**Confidence:** 2

**Summary:**

This paper analyzes theoretically some learning rates schedules commonly used in practice in ML, the cyclical learning rate and the linear decay to zero learning rate.
More precisely, it generalizes previous analyses in terms of mean-squared error to not only tackle polynomially decreasing learning rates, but also more learning rate schedules that previously lacked theoretical support, such as the linearly decaying to zero learning rate (D2Z) and cyclical learning rates.
Additionally, the results are verified experimentally.

**Questions:**

- 1. Could the authors provide maybe more details on how this work, for the convergence rate in the case of the linear schedule, compares with the work of Defazio, and whether there is any fundamental difficulty to obtain their result compared to the one in Defazio ? (even if, I agree, this work is general as it uses $\ell_p$ norms for the moments, and also, the result for linear decaying learning rate is just one result amongst many results proven in the paper so that does not put in question the rest of the paper). Especially when p=2: the result of Defazio is a convergence result in objective function, but wouldn’t one be able to obtain a result in terms of $\ell_2$ distance to optimality directly from the objective function bound, by using Assumption 2.1 at the optimum ?
- 2. Why using the p-norm in Theorem 2.1 (instead of simply the $\ell_2$ norm) ? I may have missed the motivations behind it but if it is not present in the paper I guess the authors could elaborate on why it is important to prove a result in p-norm.

**Ethical Concerns:**

["NO or VERY MINOR ethics concerns only"]

**Final Justification:**

As mentioned in the review and updated response, I am recommending accepting this paper as I think it provides interesting insights especially in the case of cyclical schedules. But my confidence score is low so I don't have a strong opinion regarding the final decision.

**Limitations:**

no specific limitation

**Paper Formatting Concerns:**

no concern

**Quality:**

3

**Strengths And Weaknesses:**

# Strengths

-  1. I think this paper makes good contributions to the literature of SGD by focusing on learning rate schedules that are recent and more used in practice.
- 2. The analysis for the linear decaying learning rate looks interesting, as it it expressed in a general $\ell_p$ norm form which looks novel compared to state of the art results for this setting such as Defazio et. al. (2023)
- 3. Perhaps more importantly the analysis for cyclical schedules looks interesting and novel since for its proof it uses tools from non-linear time series theory (from Li et. al. 2024), which is not very standard in convergence proofs in optimization (which are usually rather based on judicious refactoring of norm inequalities and using the assumptions ((local) strong convexity, etc) of the problem). The way that the results of Li et. al. are used seem novel up to my knowledge (I checked the paper from Li et. al. and did not see any proof for a cyclic schedule there).
- 4. Finally, the experiments show well the validity of the results in practice, in particular, they are able to showcase the periodicity in the standard deviations, when using a cyclic schedule.

# Weaknesses

- 1. I think it would be good for that paper to compare in slightly more details to the result from Defazio et. al. (2023), in particular, in 2.3, what do the authors mean by “its non-asymptotic convergence rate
remains unknown” ? From my understanding, the work from Defazio is non-asymptotic already ? (as it gives a bound for a given iteration number ?)
- 2. Further on this point, it seems to me at first sight that the proof in the case of linear decay, in terms of the p-norm, is somehow a standard modification of a usual proof of convergence in such setting, but using generalizations of $\ell_2$ inequalities to p-norms, but I may be wrong. If so, maybe the authors could describe in more details the fundamental difference with other usual proofs, except the use of the p-norm. (I guess in any case it can always be a mildly interesting feature to show such proof in the p-norm).

- 3. **Minor typo:** in 3.2, the reference for the Figure 1 is missing (it is written Figure ??)

---

> ### Author Rebuttal · Authors · 2025-07-31
>
> We sincerely thank the reviewer for their expansive and orderly comments, along with acknowledgment for our theoretical contribution on `Linear-D2Z` schedule, as well as our novel use of tools from non-linear time series theory for cyclical learning schedules. In the following, we address your concerns one by one. We warmly welcome any other additional suggestions that you might have. In the event you do not have any other concerns, we humbly request you to consider increasing your score.
>
> ## Weaknesses:
>
> - **I think it would be good for that paper to compare in slightly more details to the result from Defazio et. al. (2023), in particular, in 2.3, what do the authors mean by “its non-asymptotic convergence rate remains unknown”? From my understanding, the work from Defazio is non-asymptotic already? (as it gives a bound for a given iteration number?)**
>
> *Response.* Thank you very much for your comment. We strongly agree that the work from Defazio is already non-asymptotic. However, we would like to clarify that in Defazio et al. (2023), the non-asymptotic rates have been provided for the error of loss functions, see for example the convergence of $\mathbb{E}[f(x_T) - f_{\star}]$ in their Theorem 1. Differently, the primary focus of our paper is to directly provide non-asymptotic convergence for the estimates $\theta_n$ (denoted by $x_T$ in Defazio et al. (2023)), i.e., the moment convergence of $\\|\theta_n - \theta^{\star}\\|_p$  for some  $p \ge 2$. In general, the convergence rate of loss functions cannot imply the one of the estimates. While we do agree with your Question 1 that under our strong convexity assumption such result with $p = 2$ can be obtained, this leaves $p > 2$ still unexplored. We have revised this statement to make it more specific about the non-asymptotic rate for $\\|\theta_n - \theta^{\star}\\|_p$ with $p > 2$ to avoid confusion.
>
> - **Further on this point, it seems to me at first sight that the proof in the case of linear decay, in terms of the p-norm, is somehow a standard modification of a usual proof of convergence in such setting, but using generalizations of $\ell_2$ inequalities to p-norms, but I may be wrong. If so, maybe the authors could describe in more details the fundamental difference with other usual proofs, except the use of the p-norm. (I guess in any case it can always be a mildly interesting feature to show such proof in the $p$-norm).**
>
>   *Response.* We greatly appreciate your feedback. Indeed, the key of our proof for Theorem 2.2, the moment convergence of linear-decaying SGD, is not significantly different from the standard proofs but uses generalizations of the $\ell^2$-inequalities to $p$-norms. However, we would like to clarify that the main goal of our work is to provide asymptotic theory for a broad class of learning rates, and we choose to include the linear decay one due to its popularity in learning schedules, as introduced by Defazio et al. (2023). We would say the most challenging proofs are those for cyclical step sizes, where one needs to introduce new tools to study the stationarity of the sequence. According to your suggestion, we have expanded upon this discussion and added a remark in our revision.
>
> - **Minor typo: in 3.2, the reference for the Figure 1 is missing (it is written Figure ??).**
>
>   *Response.* Thank you for bringing our attention to this. The broken link in question is indeed for Figure 1, and has been fixed in the manuscript.
>
> ## Questions:
>
> - **Could the authors provide maybe more details on how this work, for the convergence rate in the case of the linear schedule, compares with the work of Defazio, and whether there is any fundamental difficulty to obtain their result compared to the one in Defazio? Especially when $p = 2$: the result of Defazio is a convergence result in objective function, but wouldn’t one be able to obtain a result in terms of $\ell_2$ distance to optimality directly from the objective function bound, by using Assumption 2.1 at the optimum?**
>
>   *Response.* As we mention in our rebuttal to your **Weakness** 1, we do agree that when $p = 2$, the strong convexity assumption together with the non-asymptotic rate of loss-functions can indicate the rate of $\\|\theta_n - \theta^*\\|_p$. However, we would like to emphasize that it is still important to derive $\ell^p$-type results given the high demand in real applications (please see our reply to your next question for examples) and the necessity of higher-moment convergence for other types of results such as almost sure convergence and convergence in distribution. Our $\ell^p$-moment convergence can serve as a useful tool for these potential future studies.
>
> - **Why using the $p$-norm in Theorem 2.1 (instead of simply the $\ell_2$ norm)? I may have missed the motivations behind it but if it is not present in the paper I guess the authors could elaborate on why it is important to prove a result in $p$-norm.**
>
>   *Response.* Thanks so much again for the great question. The $\ell^2$ convergence does not capture higher-order moments, leaving the behavior of SGD in $\ell^p$ norms unexplored. Higher-order moments, such as $\ell^p$ convergence for $p > 2$, are particularly important in applications where large deviations or outliers must be penalized more heavily. For instance:
>
>   - In financial risk management, $\ell^p$ norms are used to emphasize extreme losses in portfolio optimization and risk assessment (Rockafellar and Uryasev, 2000; Palczewski, 2018).
>   - In environmental sciences, $\ell^p$ norms are valuable in modeling extreme events such as floods or pollution spikes, where average performance is less relevant (Katz, 2002; Kratz, 2019).
>   - In reinforcement learning, particularly in high-risk decision-making scenarios, penalizing large errors using $\ell^p$ norms ensures that models learn to avoid catastrophic mistakes more effectively (Moody and Saffell, 2001; Tamar et al., 2015).
>
>   Moreover, if one aims to derive other types of non-asymptotic convergence rates such as the Berry–Esseen bound in Gaussian approximation, then higher-order moments would be required. We will add this discussion to our revision for clearer motivation.

---

> > ### Comment · Reviewer_8TRU · 2025-08-06
> > **response to authors**
> >
> > Dear authors, thanks a lot for your answer, your rebuttal addresses my concerns and as such I will maintain my positive review of your paper.

---

### Official Review · Reviewer_gHWv · 2025-07-03

**Clarity:** 4
**Significance:** 3
**Originality:** 4
**Rating:** 4
**Confidence:** 3

**Summary:**

This paper addresses the gap between the well-established theoretical analyses of stochastic gradient descent (SGD) with polynomially decaying step-sizes and the practical use of alternative learning rate schedules such as cyclical learning rates and linearly decaying to zero (Linear-D2Z), which have been widely adopted in deep learning but lack comprehensive theoretical understanding. The authors develop a unified framework that provides non-asymptotic mean-squared error bounds for SGD iterates under arbitrary learning rate schedules, generalizing classical results and explicitly capturing the influence of initial conditions and step-size sequences. They offer the first rigorous convergence characterization for the popular Linear-D2Z schedule and introduce novel asymptotic theory for cyclical learning rates, showing that SGD with cyclical rates converges to a cyclo-stationary distribution exhibiting periodic behavior, in contrast to the stationary distribution convergence of constant step-size SGD. Numerical experiments validate these theoretical insights, demonstrating fast early convergence and low final error for linearly decaying schedules and cyclical variance patterns for cosine schedules.

**Questions:**

Please see above.

**Ethical Concerns:**

["NO or VERY MINOR ethics concerns only"]

**Final Justification:**

I stand with my initial judgement - after reading other reviews and the additional experiments, I did not find much that changed/improved my initial opinion on the paper. I still lean towards acceptance.

**Limitations:**

Yes

**Paper Formatting Concerns:**

The github repository is not anonymized...

**Quality:**

3

**Strengths And Weaknesses:**

I think in general the paper is well written, the motivation is clear and the contribution too.

One of my main issues with the paper is that I find the title and some parts of the abstract and introduction a bit misleading. I do not want to diminish the contribution of the paper, but they way it is presented, especially in the title for a "general learning-rate", the reader might be misled thinking that the proposed framework generalizes all the previous works so far, or is insightful for them. However this is not the case, as for constant learning rate, as the authors point out, later on, their Theorem 2.1 only yields a O(1) bound, which does not seem useful. I do not have an issue with this per se, but with the way the framework is presented and encourage the authors to change this.

The authors make several assumptions which I do not seem to be too restricting and are standard assumptions in works like this, and they further give comments on these assumptions and relaxations in case where they are possible which I appreciated.

In Remark 2.4 the authors say that some of the ηi’s maybe taken to be large which helps in faster convergence. This can also be seen empirically in Figure 4. Could the authors be a bit more specific to how they define "large" and provide a bit more clarification on this? A histogram for the norms might be interesting. Another thing that I observe in the is: in first example in Figure 4 with 0.1 as the multiplicative constant, the norms of these fluctuations seem to start to be larger and larger as the process progresses. Why did the authors consider 10^4 iterations and not more? For the smaller constant multipliers it seems that the behavior does not change but for the case 0.1 I think it would be useful to further run the simulation.

Although the paper is of theoretical nature, it would be beneficial and I would have been keen to give a higher score if the authors have performed more extensive experimental analysis, for example using simple networks as LeNet or AlexNet would suffice on simple datasets like MNIST or CIFAR-10 that do not require intensive computational resources. I think this would make the paper stronger.

Also, can the authors repeat their experiments with initializations following a zero-centered Gaussian distribution? I think this initialization is, at least from my experience in similar works, a lot more standard. I am not sure why did the authors decide to fix it throughout all experiments and it raised slight confusion, although I believe that the results and findings should not change.

---

> ### Author Rebuttal · Authors · 2025-07-31
>
> We sincerely thank the reviewer for their comprehensive appraisal of our paper, in particularly highlighting our theoretical contribution on `Linear-D2Z`, detailing the first rigorous convergence characterization for that scheme, as well as for cyclical learning rate schedules. We warmly welcome any other additional suggestions that might improve the paper, and wholeheartedly appreciate your willingness to increase the score given more experimental evidence. In particular, we have taken the opportunity of this rebuttal to demonstrate some experimental results on MNIST dataset. However, before discussing that directly, let us address your concerns one by one.
>
> ## Weaknesses:
>
> - **One of my main issues with the paper is that I find the title and some parts of the abstract and introduction a bit misleading. I do not want to diminish the contribution of the paper, but the way it is presented, especially in the title for a "general learning-rate", the reader might be misled thinking that the proposed framework generalizes all the previous works so far, or is insightful for them. However this is not the case, as for constant learning rate, as the authors point out, later on, their Theorem 2.1 only yields a O(1) bound, which does not seem useful. I do not have an issue with this per se, but with the way the framework is presented and encourage the authors to change this.**
>
>   *Response.* We greatly appreciate your suggestion. In the revision, we will change the title to **"Asymptotic theory of SGD for a broad class of learning rates"** to avoid confusion. For the other parts mentioning "general learning rates", especially the ones in the abstract and introduction, we will revise the statements accordingly.
>
> - **In Remark 2.4 the authors say that some of the $\eta_i$’s may be taken to be large which helps in faster convergence. This can also be seen empirically in Figure 4. Could the authors be a bit more specific to how they define "large" and provide a bit more clarification on this? A histogram for the norms might be interesting.**
>
>   *Response.* We agree with your concern — "large" is relative, and should be calibrated according to the problem at hand. This is best understood as relatively large and relatively small learning rates within the same cyclical schedule. In this paper, we've opted to demonstrate this with an easy to understand example — the cosine learning rate schedule with period 3, resulting in one $\eta_i$ in each cycle of three being four times as large as the others.
>
> - **Another thing that I observe is: in first example in Figure 4 with 0.1 as the multiplicative constant, the norms of these fluctuations seem to start to be larger and larger as the process progresses. Why did the authors consider $10^4$ iterations and not more? For the smaller constant multipliers it seems that the behavior does not change but for the case 0.1 I think it would be useful to further run the simulation.**
>
>   *Response.* To address your concern, we present the results of another simulation, with $n = 10^5$, $\theta^* = (0,0)$ and $\theta_0$ initialized randomly. The other settings remain as they were in the paper:
>
>   $$
>   y_i = \theta^{(0)} + \theta^{(1)} x_i + \varepsilon_i,\quad \varepsilon_i \sim \mathcal{N}(0,1)\text{ i.i.d.},\quad \theta^* = (\theta^{(0)}, \theta^{(1)})^\top \in \mathbb{R}^2
>   $$
>
>   where $(x_i, y_i) \in \mathbb{R}^2$ denotes the observed data and $\theta^* \in \mathbb{R}^2$ is the unknown parameter. Results are averaged over 500 SGD runs.
>
>   **Table: Final and Minimum MSE for Cosine Learning Rate Schedules on Simulation Data**
>
>   | Schedule                          | Final MSE | Min MSE | Final SE SD | Mean SE SD (last 100) | Iteration Range |
>   |----------------------------------|-----------|---------|--------------|------------------------|------------------|
>   | 0.1 · (1 + cos(2$\pi$ t / 3))         | 0.1713    | 0.1103  | 0.1743       | 0.2334                 | 500–5000         |
>   | 0.05 · (1 + cos(2$\pi$ t / 3))        | 0.0805    | 0.0484  | 0.0777       | 0.0912                 | 500–5000         |
>   | 0.01 · (1 + cos(2$\pi$ t / 3))        | 0.0158    | 0.0101  | 0.0136       | 0.0144                 | 500–5000         |
>
> - **Although the paper is of theoretical nature, it would be beneficial and I would have been keen to give a higher score if the authors had performed more extensive experimental analysis. For example, using simple networks such as LeNet or AlexNet on simple datasets like MNIST or CIFAR-10 that do not require intensive computational resources. I think this would make the paper stronger.**
>
>   *Response.* To address your concerns regarding the scope of our empirical evaluation, we conducted additional experiments on the MNIST dataset using a high-dimensional classification task. Specifically, we trained a multiclass logistic regression model via stochastic gradient descent (SGD) under both the cosine and `Linear-D2Z` learning rate schedules.
>
>   Each MNIST image is flattened into a vector $x \in \mathbb{R}^{784}$ and paired with a one-hot encoded label $y \in \{0,1\}^{10}$. Given this input-target pair, we minimize the sigmoid loss $\mathcal{L}(x, y; \theta)$, where $\theta \in \mathbb{R}^{784 \times 10}$ denotes the model parameters. This setup is equivalent to minimizing a sum of binary logistic regression losses across classes in a one-vs-rest fashion.
>
>   For the cosine schedule $\eta_t = \eta_0 (1 + \cos(2\pi t / 3))$, we ran SGD for $n = 5000$ iterations, anticipating convergence to a cyclostationary distribution. The outcome was indeed cyclostationary in nature, similarly to what was observed in Figure 4.
>
>   For the `Linear-D2Z` schedule $\eta_t = \eta_0 (1 - t/n)$, we ran SGD in increments of 500 iterations, from $n = 500$ to $n = 5000$. Performance was evaluated in terms of both the average sigmoid loss and the classification accuracy (i.e., the proportion of correctly classified digits under $\arg\max_j \theta^\top x$).
>
>   **Table: Sigmoid Loss Estimate for the `Linear-D2Z` Schedule on the MNIST Dataset**
>
>   | Number of Iterations | MSE    | Standard Deviation |
>   |----------------------|--------|--------------------|
>   | 500                  | 0.0059 | 0.0046             |
>   | 1000                 | 0.0042 | 0.0020             |
>   | 1500                 | 0.0041 | 0.0015             |
>   | 2000                 | 0.0034 | 0.0018             |
>   | 2500                 | 0.0034 | 0.0013             |
>   | 3000                 | 0.0035 | 0.0012             |
>   | 3500                 | 0.0033 | 0.0011             |
>   | 4000                 | 0.0032 | 0.0011             |
>   | 4500                 | 0.0032 | 0.0012             |
>   | 5000                 | 0.0031 | 0.0011             |
>
> - **Also, can the authors repeat their experiments with initializations following a zero-centered Gaussian distribution? I think this initialization is, at least from my experience in similar works, a lot more standard. I am not sure why did the authors decide to fix it throughout all experiments and it raised slight confusion, although I believe that the results and findings should not change.**
>
>   *Response.* See the response to your concern with Figure 4.

---

> > ### Comment · Reviewer_gHWv · 2025-08-03
> >
> > I thank the authors for changing the paper title, as well as responding to my other questions. Regarding the MNIST Dataset example, it would be much more beneficial if the authors can compare performance to other schedulers: the observed cyclostationarity is indeed a good sign but I feel like it would be good to know how does the scheduler compare with other schedulers.

---

> ### Author Response · Authors · 2025-08-05
> **Comparing cosine  and constant schedules**
>
> We sincerely appreciate the reviewer for the suggestion. Regarding the MNIST Dataset example, we have added the comparison of the cosine schedule and the constant learning rate, another popular choice of the learning schedule.
>
> As mentioned in our previous comment, we trained a multiclass logistic regression model via stochastic gradient descent (SGD) under both the cosine and constant learning rate schedules. Each MNIST image is flattened into a vector $x \in \mathbb{R}^{784}$ and paired with a one-hot encoded label $y \in \\{0,1\\}^{10}$. Given this input-target pair, we minimize the sigmoid loss $\mathcal{L}(x, y; \theta)$, where $\theta \in \mathbb{R}^{784 \times 10}$ denotes the model parameters. This setup is equivalent to minimizing a sum of binary logistic regression losses across classes in a one-vs-rest fashion.
>
> For the cosine schedule $\eta_t = \eta_0 (1 + \cos(2\pi t / 3))$, we ran SGD for $n = 5000$ iterations. For the constant learning rate, we fixed $\eta_t = \eta_0$ to make comparisons. In both learning schedules, we considered $\eta_0=0.001$ for improved precision. We report the average loss for this process through steps 1-1000, 2001-3000 and 4001-5000 in the table below.
>
> **Table: Sigmoid loss estimates for cosine and constant learning rate schedules on the MNIST dataset**
>
> | $\eta_0$ | Schedule Type | Loss (1–1000) | Loss (2001–3000) | Loss (4001–5000) |
> |---------:|---------------|---------------|------------------|------------------|
> | 0.001    | Cosine        | 1.9144        | 1.0986           | 0.8337           |
> |          | Constant      | 1.9254        | 1.0775           | 0.8626           |
>
> We observe that with a small $\eta_0$, the terminal average loss for cosine and constant learning rate schedules with the same $\eta_0$ becomes near indistinguishable, even slightly favoring the cosine schedule.

---

### Official Review · Reviewer_WHTk · 2025-07-21

**Clarity:** 2
**Significance:** 2
**Originality:** 3
**Rating:** 4
**Confidence:** 4

**Summary:**

This paper studies how stochastic gradient descent (SGD) behaves when using different kinds of learning rate schedules—especially ones commonly used in practice, like linearly decaying learning rates and cyclical ones. These types of schedules are widely used in training deep learning models but haven’t been fully studied in theory. The authors develop a general framework to analyze these learning rates, providing both theoretical results and experiments. The main contributions include error bounds for general step sizes (Theorem 2.1), analysis of linearly decaying learning rates (Theorem 2.2), and new results about cyclical learning rates (Theorem 2.4), showing that they converge to a repeating (cyclostationary) pattern.

**Questions:**

Please comment about indicate weaknesses. Does convexity assumptions could be relaxed?

**Ethical Concerns:**

["NO or VERY MINOR ethics concerns only"]

**Final Justification:**

Due to the promise to improve a paper I increase my score

**Quality:**

3

**Strengths And Weaknesses:**

Strengths
The new theroretical analysis of linear decay to zero SGD.

Weakness
In the introduction, the paper does not explain what $\eta_t$ and $\eta_{n,t}$ represent, which may confuse readers unfamiliar with the notation. The schedule $\eta_t = \eta(1 - t/n)$ is introduced without much context or justification. This type of schedule assumes that the total number of iterations $n$ is known in advance, which is often unrealistic in practice—especially for stochastic algorithms where the convergence time is not fixed and may vary depending on the data or noise. Additionally, the paper does not cite some relevant prior work.
Assumption 2.3 seems to be more restrictive than usual assumption on boundedness of conditional expectation. In this form this assumption requires that noisy gradient could not depend on past trajectory.

I will reconsider my opinion based on author rebuttal.

---

> ### Author Rebuttal · Authors · 2025-07-31
>
> We sincerely thank the reviewer for their detailed and pertinent comments, along with encouraging words for our theoretical contribution on `Linear-D2Z` schedule. In the following, we address your concerns one by one. We warmly welcome any other additional suggestions to improve this paper, and wholeheartedly appreciate your decision to reconsider the score based on this rebuttal.
>
> ## Weaknesses:
>
> - **In the introduction, the paper does not explain what $\eta_t$ and $\eta_{n,t}$ represent, which may confuse readers unfamiliar with the notation.**
>
>   *Response.* We appreciate the note on clarity — these both refer to the very same $\eta_t = \eta(1 - t/n)$, with or without the additional $n$ in the subscript whenever it may be necessary to specify that this depends on a particular value of $n$. Notes clarifying this have been added to the manuscript.
>
> - **The schedule $\eta_t = \eta(1 - t/n)$ is introduced without much context or justification. This type of schedule assumes that the total number of iterations $n$ is known in advance, which is often unrealistic in practice—especially for stochastic algorithms where the convergence time is not fixed and may vary depending on the data or noise. Additionally, the paper does not cite some relevant prior work.**
>
>   *Response.* We thank you for raising this point. As the reviewer *gHWv* also mentioned, the `Linear-D2Z` schedule is widely explored in the literature, in particular in AdamW training (Loshchilov and Hutter, *Decoupled Weight Decay Regularization*, ICLR 2019), as well as in Bregsma et al., *Straight to Zero: Why Linearly Decaying the Learning Rate to Zero Works Best for LLMs*, ICLR 2025, and in Defazio et al., *Optimal Linear Decay Learning Rate Schedules and Further Refinements*, arXiv preprint, 2023, and references therein. The latter has already been cited, and the former has been added to the manuscript for improved clarity.
>
>   Apart from citing these papers and others, we explain the utility of this schedule — one can view its early stages as akin to a fixed learning rate, allowing for fast approach to the solution, and its latter stages decay faster than a polynomial rate, allowing for improved accuracy. Furthermore, 1-batch single epoch SGD is a common approach, and in line with the theoretical work presented in every theoretical step in this paper. Under this framework, the number of steps $n$ is determined by the length of the dataset. One can adapt $n$ if this does not lead to a satisfactory empirical result (i.e., insufficient information on long-term behavior), but this has not been the case here.
>
> - **Assumption 2.3 seems to be more restrictive than the usual assumption on boundedness of conditional expectation. In this form, this assumption requires that noisy gradient could not depend on past trajectory.**
>
>   *Response.* Indeed, by Jensen's inequality, our Assumption 2.3 implies the usual assumption:
>
>   $$
>   |\nabla F(x) - \nabla F(y)| \leq L|x - y|, \text{ for all } x, y; \quad \mathbb{E}[|\nabla F(\theta_{t-1}) - \nabla f(\theta_{t-1}, \xi_t)|^2 \mid \mathcal{F}_{t-1}] < M_2.
>   $$
>
>   The Lipschitz condition is very classical in the asymptotic analysis of SGD (see Assumption 4.3, Polyak and Juditsky, *Acceleration of Stochastic Approximation by Averaging*, SIAM Optimization, 1992). We note that our proof of Lemma B.2 (the main result behind Theorem 2.1) goes through verbatim with this slightly weaker assumption. Additionally, we point out that our sampling process for $\xi_t$ is i.i.d., and thus the noisy gradient $\nabla f(\theta_{t-1}, \xi_t)$ depends on the past trajectory only through $\theta_{t-1}$, similar to a Markov chain. We gladly welcome any further comments from your side if you think further clarifications are required.
>
> ## Questions:
>
> **Does convexity assumptions could be relaxed?**
>
> *Response.* We thank the reviewer for raising this pertinent question. If there are only finitely many local minima, our techniques along with a *local strong convexity* assumption can yield the corresponding results conditional on in-probability convergence. See, for example, Zhong et al., *Online Bootstrap Inference with Non-Convex Stochastic Gradient Descent Estimator*, Preprint 2023.
>
> Unfortunately, for non-convex regimes with uncountably many local minima, the error bounds may not be obtained in an $\mathcal{L}_2$ sense, and it might be more convenient to look at regret bounds or convergence of $\nabla F(\theta_t)$. As the reviewer *H11d* points out, PL inequality may be a more appropriate assumption there. For general choices of step sizes, this indeed represents an exciting research direction, and we believe — despite the significant non-triviality — the techniques developed here should be useful there too.

---

> > ### Author Response · Authors · 2025-08-06
> > **Appreciation for Your Review – Follow-Up on Responses**
> >
> > Dear Reviewer,
> >
> > Thank you again for your time and thoughtful feedback on our submission. We hope you have had a chance to look over our responses during the rebuttal phase.
> >
> > If there are any remaining questions or clarifications we can provide, we would be more than happy to do so. We also hope our responses addressed your concerns fully. If that is the case, we would greatly appreciate your consideration in updating your score.
> >
> > We are truly grateful for your engagement and for helping strengthen our work.

---

> > ### Comment · Reviewer_WHTk · 2025-08-07
> >
> > Thank you for your answer. I agree that the scheme with a fixed length is present in the literature. It will be very good if you are able to proof main results under weaker condition.

---

> ### Author Response · Authors · 2025-08-07
> **Proofs under weaker conditions**
>
> Thank you for raising these concerns about our conditions. As already mentioned in our rebuttal, the results can be easily extendable to accommodate much weaker, ``local concordance property''. For example, logistic regression satisfies the local concordance property, but is *not* strongly convex. We have edited our proofs to reflect their viability under such weaker conditions, and show the key change in the following.
>
> A recurring theme of our proofs is to show that
>
> $$
> |\theta - \theta^\star -\eta \nabla F(\theta)|^2 \leq (1-\eta c)|\theta-\theta^\star|^2  \quad \text{for some $c>0$, $\theta \in \mathbb{R}^d$.}
> $$
>
> We highlight the different arguments leading up to the above inequality, leveraging local strong concordance.
>
>
> ## Proof of the inequality via local strong concordance
>
> Fix $\theta \in \mathbb{R}^d$, and choose $\phi(u) = F(\theta^\star+ u(\theta- \theta^\star))$ for $u\in[0,1]$. Note that $\phi^{\prime\prime}(0) \geq  \mu^{\star} |\theta-\theta^\star|^2$. From local strong concordance, one directly has
>
> $$
> \phi^{\prime \prime}(u) \geq  \phi^{\prime \prime}(0) \exp(-C|\theta- \theta^\star|u),
> $$
>
> and therefore, recalling $|x|\leq R$,
>
> $$
> \begin{align*}
> (\theta - \theta^\star)^\top \nabla F(\theta) &= \phi^\prime(1)- \phi^\prime(0) \\
> &\geq  \mu^{\star} |\theta-\theta^\star|^2 \int_0^1 \exp(-C|\theta- \theta^\star|u) \,\mathrm{d}u \\
> &=  \mu^{\star} |\theta-\theta^\star|^2 \frac{1 - \exp(-C|\theta- \theta^\star|)}{C|\theta- \theta^\star|} \\
> &\geq  \mu^{\star} C\exp(-R) |\theta-\theta^\star|^2 ,
> \end{align*}
> $$
>
> which immediately can be applied to the previous quadratic form to deduce the leading term contraction.
>
> ## Non-convexity
>
> As for the $\mathcal L_2$ error-bound proofs (Theorems 2.1 and 2.2)  under potential non-convexity, if there are only finitely many local minima, our techniques along with a local strong convexity assumption yield the corresponding results conditional on in-probability convergence. This is again, mostly, a trivial extension of our techniques, and we have added a remark after Theorem 2.1 in this regard.  See, for example, Zhong et al., Online Bootstrap Inference with Non-Convex Stochastic Gradient Descent Estimator, Preprint 2023.
>
> For non-convex regimes with uncountably many local minima, we have already described the main technical issues in our rebuttal , under the Question **Does convexity assumptions could be relaxed?**. We reproduce it here for convenience.
>
> > "Unfortunately, for non-convex regimes with uncountably many local minima, the error bounds may not be obtained in an $\mathcal L_2$ sense, and it might be more convenient to look at regret bounds or convergence of $\nabla F(\theta_t)$. As the reviewer *H11d* points out, PL inequality may be a more appropriate assumption there. For general choices of step sizes, this indeed represents an exciting research direction, and we believe — despite the significant non-triviality — the techniques developed here should be useful there too."
>
> Finally, for the Theorem 2.4, under potentially multiple local minima, we again expect some sort of conditional convergence to hold, but it is severely technically challenging not least due to the measure theoretic issues while combining the cyclo-stationary convergence with each corresponding local minima. This represents an exciting research direction which we are determined to pursue.
>
> We hope that our explanation has fully addressed your concerns.  We warmly welcome any other comments or feedbacks to improve this paper. If you find our explanation satisfactory, we request you to consider a higher score.

---

> > ### Comment · Reviewer_WHTk · 2025-08-08
> >
> > Thank you for writing the proof here. I hope that it will be inculded in finakl version

---

> > > ### Author Response · Authors · 2025-08-09
> > >
> > > We thank the reviewer for initiating this insightful discussion, and for taking the time to review our paper in detail. We will make sure to include this proof in our final version. Since the discussion period is ending, we kindly request you to consider a higher score if you found our explanation adequate.

---

### Note · Authors · 2025-08-13

We sincerely appreciate the valuable suggestions from the reviewers, which have been instrumental in strengthening our work. The reviewers found the paper clearly written, well-motivated, and significant in both its results and potential impact. In particular, they appreciated the following key contributions:

- A *first-of-its-kind* general framework for understanding non-asymptotic error bounds for a broad class of learning rate schedules (Thm. 2.1), including novel analysis of alternatives such as linear-D2Z and cyclical schedules, beyond the constant or polynomially decaying cases.
- The asymptotic convergence to a cyclostationary distribution for the cyclical schedule (Thm. 2.4). We gratefully found the reviewers in consensus about our result being the *first* such theoretical result for cyclical schedules.
- General non-asymptotic $\ell_p$ error bounds for the linear-D2Z learning rate schedule.
- Novel proof techniques from nonlinear time series theory for cyclical schedules.
- Extensive numerical experiments validating theoretical predictions for each schedule.

The reviewers also raised valuable suggestions to improve our work. Main concerns include:

- Comparison of our assumptions to commonly used ones for similar problems and possible generalization to non-convex settings.
- Explaining our reasoning behind the choice of $n$ in the linear-D2Z schedule.
- Expanding our empirical study to other initializations and larger sample sizes, as well as to existing benchmark datasets such as MNIST.
- Explaining the novelty in our proof techniques, particularly those generalizing to $\ell_p$ error.

Through our rebuttals and discussions with the reviewers, we have resolved each point with clarifications, additional technical proof details, and outlined further experiments and examples.

In the camera-ready version, we will incorporate all revisions described above. Finally, we would like to extend our heartfelt thanks to the reviewers, the AC and the SAC for their dedicated time, insightful expertise and thoughtful engagement with our work. Your feedback has been instrumental in clarifying subtle points, refining our arguments and strengthening empirical validation. These improvements have significantly enhanced the clarity, rigor and impact of the paper. We sincerely hope that our paper will be evaluated favorably for its originality, comprehensive theory and experiments and substantial revisions made in response to the reviews.

---

### Decision · Program_Chairs · 2025-09-17

**Decision:**

Accept (poster)

**Comment:**

This paper analyses the convergence of SGD with some learning rate schedules commonly used in practice such as linear decay to zero and cyclical schedules. The aim was to fill a gap between these widely learning rate schedules and existing theory, which focuses mostly on polynomially decaying or constant step sizes. The main contributions include general non-asymptotic convergence bounds applicable to a wide range of schedules, explicit rates for linear decay, and the first rigorous results on the limiting distribution under cyclical learning rates using ideas from nonlinear time series analysis. Reviewers agreed the theory is correct and original, and that the results are well-motivated and may have substantial impact.

The main limitation is that the experiments could be more extensive. While some technical assumptions (e.g., strong convexity, etc.) limit direct applicability to deep models, the authors provided an extension of their result under weaker assumptions, and promised further improvement and clarification to the presentation. It is critical that the authors incorporate all promised changes into the camera-ready version.